# CAN LARGE LANGUAGE MODELS UNDERSTAND SYMBOLIC GRAPHICS PROGRAMS?

Zeju Qiu[1,†]   Weiyang Liu[1,2,†,*]   Haiwen Feng[1,†]   Zhen Liu[1,‡]   Tim Z. Xiao[1,‡]   Katherine M. Collins[2,‡]
Joshua B. Tenenbaum[3]   Adrian Weller[2]   Michael J. Black[1]   Bernhard Schölkopf[1]
[1]Max Planck Institute for Intelligent Systems, Tübingen   [2]University of Cambridge   [3]MIT
[†]Joint first author   [‡]Joint second author   [*]Project lead   `sgp-bench.github.io`

## ABSTRACT

Against the backdrop of enthusiasm for large language models (LLMs), there is a growing need to scientifically assess their capabilities and shortcomings. This is nontrivial in part because it is difficult to find tasks which the models have not encountered during training. Utilizing symbolic graphics programs, we propose a domain well-suited to test multiple spatial-semantic reasoning skills of LLMs. Popular in computer graphics, these programs procedurally generate visual data. While LLMs exhibit impressive skills in general program synthesis and analysis, symbolic graphics programs offer a new layer of evaluation: they allow us to test an LLM's ability to answer semantic questions about the images or 3D geometries without a vision encoder. To semantically understand the symbolic programs, LLMs would need to possess the ability to "imagine" and reason how the corresponding graphics content would look with only the symbolic description of the local curvatures and strokes. We use this task to evaluate LLMs by creating a large benchmark for the semantic visual understanding of symbolic graphics programs, built procedurally with minimal human effort. Particular emphasis is placed on transformations of images that leave the image level semantics invariant while introducing significant changes to the underlying program. We evaluate commercial and open-source LLMs on our benchmark to assess their ability to reason about visual output of programs, finding that LLMs considered stronger at reasoning generally perform better. Lastly, we introduce a novel method to improve this ability – *Symbolic Instruction Tuning* (SIT), in which the LLM is finetuned with pre-collected instruction data on symbolic graphics programs. Interestingly, we find that SIT not only improves LLM's understanding on symbolic programs, but it also improves general reasoning ability on various other benchmarks.

## 1 INTRODUCTION

What are large language models (LLMs) capable of? Recent studies [5, 58] have shown that LLMs are able to generate generic computer programs, indicating a degree of pragmatic understanding of the symbolic structure of programs. Motivated by this progress, we focus on another important family of computer programs, called symbolic graphics programs, where a graphics content (*e.g.*, image, 3D asset) can be generated by running a program. We are interested in the following question: *Can large language models "understand" symbolic graphics programs?*

Before trying to answer this question, we start by defining what we consider "understanding" of symbolic graphics programs, in the context of this work. Because a (deterministic) graphics program can be uniquely rendered to an image (the graphics programs we consider here), we characterize LLMs' understanding of the graphics program as the semantic understanding of the corresponding rendered image. More specifically, we approximate such a semantic visual understanding by the ability to correctly answer semantic questions only based on the raw program input. These semantic questions are generated based on the rendered image, such that they are easy to answer given the image and yet challenging given only the program as text prompt. Guided by this insight, we propose a generic pipeline for creating benchmarks that can evaluate this particular ability of LLMs to understand symbolic graphics programs, while requiring minimal human effort. While we of course recognize that there are other elements of visual reasoning that characterize understanding in humans, and that we ought to evaluate in machine intelligence, we believe that our benchmark provides insight into one element of "understanding" of symbolic graphics programs that helps assay what current LLMs are (and are not) capable of.

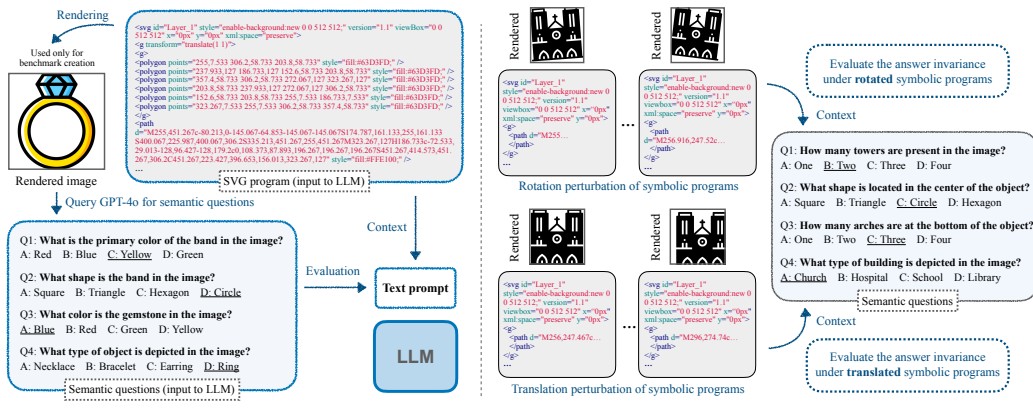

Figure 1: Our benchmark assesses LLMs' understanding of symbolic graphics programs in semantic understanding and prediction consistency. Note that the LLM can only see symbolic graphics programs and the corresponding questions. The rendered images are not input to the LLM.

The semantic understanding of symbolic graphics programs is particularly interesting in the following aspects. First, such a semantic understanding may necessitate a form of "visual imagination" ability that enables the LLM to "imagine" how the produced graphics content visually looks like. If a LLM can perfectly answer every possible semantic question of a symbolic graphics program (with sufficient knowledge, the graphics content can be reconstructed), then we can say that such a LLM may have a good "visual imagination" of this program. Second, symbolic programs represent a procedural way to generate the graphics content, and hence the semantic understanding also requires the long-range sequential reasoning of the program. The order of the symbolic operations may substantially affect its semantic meaning, making the problem quite challenging. Third, many semantic questions involve an accurate grounding of semantic components, and such a grounding in the symbolic program requires a fine-grained understanding of the program structure. This motivates us to study whether LLMs have the ability to semantically understand symbolic graphics programs, and furthermore, how to improve this ability. In general, correctly answering semantic questions about symbolic graphics programs requires a combination of multiple sophisticated reasoning abilities from LLMs, which therefore makes the task of symbolic graphics program understanding an ideal benchmark to contribute towards evaluating the holistic reasoning capabilities of LLMs. Reasoning over symbolic programs is of particular interest from a cognitive perspective as well [103, 84, 25]. To what extent LLMs can operate over such a representation with rich structures remains an open problem.

Motivated by the significance of symbolic graphics program understanding, we build a benchmark, called *SGP-Bench*, for two important variants of symbolic graphics programs: scalable vector graphics (SVG) as a generic language for representing 2D vector graphics, and customized computer-aided design (CAD) as a domain-specific language (DSL) for representing 2D/3D objects. Our benchmark consists of two types of evaluations. (1) *Semantic understanding*: We construct a number of semantic questions (*i.e.*, multiple-choice questions with 4 options) from a set of images (from multiple different categories). These questions are fed to LLMs along with the symbolic program to evaluate the semantic understanding. (2) *Semantic consistency*: To evaluate the robustness of LLM's semantic understanding, we perform random translation and rotation to the original symbolic programs and then test the same semantic questions based on the perturbed programs. We evaluate the consistency of the answers from LLMs using these perturbed symbolic programs with identical semantic meaning. This evaluation can also help lower the possibility of test data leakage, because the randomly perturbed programs are unlikely to be seen during pretraining. An overview of SGP-Bench is given in Figure 1. We further validate our automated labels via a human study (see Appendix B).

In addition to performing evaluation under the common in-context setting wherein LLMs are used "out-of-the-box" and not finetuned, we also evaluate whether finetuning LLMs on a curated dataset can boost performance. To this end, we propose *Symbolic Instruction Tuning* (SIT). The key idea is to collect an instruction dataset based on the rendered images. Because the semantic questions of interest are usually easy to answer from the visual input, we take advantage of the rendered images (that correspond to symbolic programs) and query a powerful language-vision model (*e.g.*, GPT-4o) for detailed captioning. This leads to a scalable way to collect an instruction dataset for symbolic graphics programs. Then, we simply finetune open-source LLMs on this dataset. Our experiments demonstrate that SIT can improve a model's semantic understanding of symbolic programs, and more importantly, its general reasoning ability. Our contributions are summarized below:

- We introduce a new task of symbolic graphics program understanding and propose a generic yet highly scalable benchmark creation pipeline for this task.

- We build a large benchmark, SGP-Bench, for comprehensively evaluating LLM's semantic understanding and consistency of symbolic graphics programs. In SGP-Bench, we consider two types of symbolic graphics programs: SVG for 2D vector graphics and CAD for 2D/3D objects.

- To improve the symbolic program understanding, we collect an instruction-following dataset and propose symbolic instruction tuning, which can also improve general reasoning performance.

- Finally, we introduce a symbolic MNIST dataset where the symbolic program is so challenging for LLMs to understand that GPT-4o can only achieve a chance-level performance, while the rendered image is easily recognizable by humans.

## 2 SEMANTIC UNDERSTANDING OF SYMBOLIC GRAPHICS PROGRAMS

We introduce the task of semantic symbolic graphics program understanding. Our goal is to assess to what extent a LLM is able to "understand" a symbolic graphics program, which may begin to belie some latent capability to "visually imagine". Specifically, we leverage the correspondence between deterministic symbolic graphics programs and rendered images, and then we characterize the understanding of symbolic graphics programs as the semantic understanding of the corresponding rendered image. To do so, we

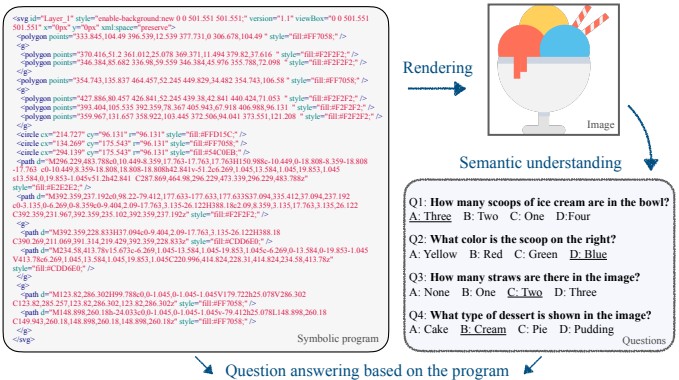

Figure 2: Illustration of the symbolic graphics program understanding task.

use the performance on question-answering to evaluate the semantic understanding of images. The same set of questions, along with the corresponding symbolic graphics programs, are then used to evaluate the symbolic program understanding of LLMs (the rendered image will not be used here). Figure 2 gives an illustration of symbolic graphics program understanding. The intuition behind this evaluation is that, if an LLM has a good sense of the symbolic graphics and implicit de-rendering, then the LLM should have a rough understanding about its rendered image such that it is able to answer arbitrary semantic questions about the rendered image.

Symbolic graphics program understanding can be viewed as a form of visual question answering in the sense that visual input is represented by a symbolic program representation. Compared to current vision-language models [56, 55, 121] that encodes images with a text-aligned encoder [76], we consider the case where the visual input are encoded by a symbolic program that can exactly recover the graphics content. From this perspective, our task aims to uncover the potential of using symbolic programs as a representation to perform visual reasoning.

## 3 WHY IS UNDERSTANDING SYMBOLIC GRAPHICS PROGRAMS INTERESTING?

To showcase why semantic understanding of symbolic graphics problems require multiple sophisticated reasoning abilities, we

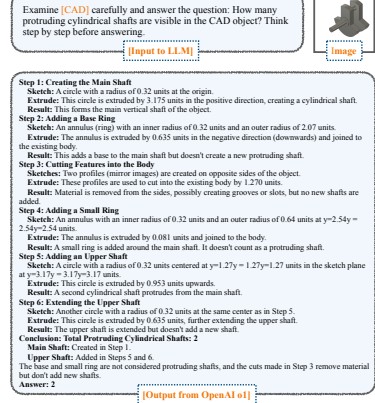

Figure 3: A qualitative example of CAD reasoning.

provide a few qualitative examples of LLM's output in Figure 3 and Figure 4. Figure 3 shows a qualitative example of how OpenAI-o1 reasons over the CAD program. The reasoning process is highly nontrivial, as it requires multiple reasoning abilities, such as numeric perception, spatial reasoning, geometric understanding, long-range planning and common sense. This is also part of the reason that SGP-Bench can well evaluate the general reasoning ability of LLMs. In Figure 4, we query different LLMs (from weak to strong: Llama-3.1-8B, Llama-3.1-70B, OpenAI-o1) by asking

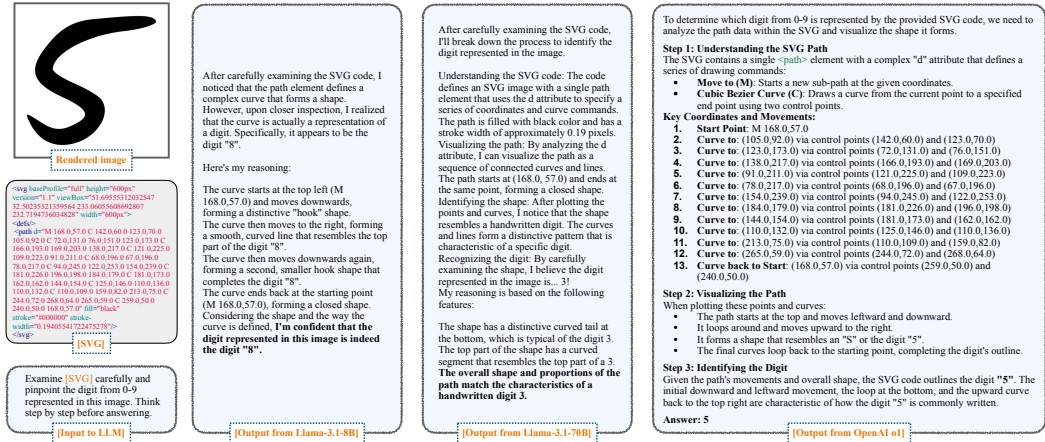

Figure 4: Qualitative examples of how LLMs reason over the symbolic program and obtain their answers.

which digit this given SVG program represents. We can observe that all LLMs start to reason from the low-level curves and gradually build up its understanding from local components to semantic elements. Specifically, LLMs understand the symbolic program through line-by-line reasoning and then combine the results into an overall semantic understanding. This process is intriguing because it shows that LLMs understand symbolic programs through reasoning rather than memorization. More interestingly, the more powerful LLMs, *e.g.*, OpenAI-o1, show better general understanding of the symbolic program and its fine-grained grounding, which is consistent with the results on SGP-Bench.

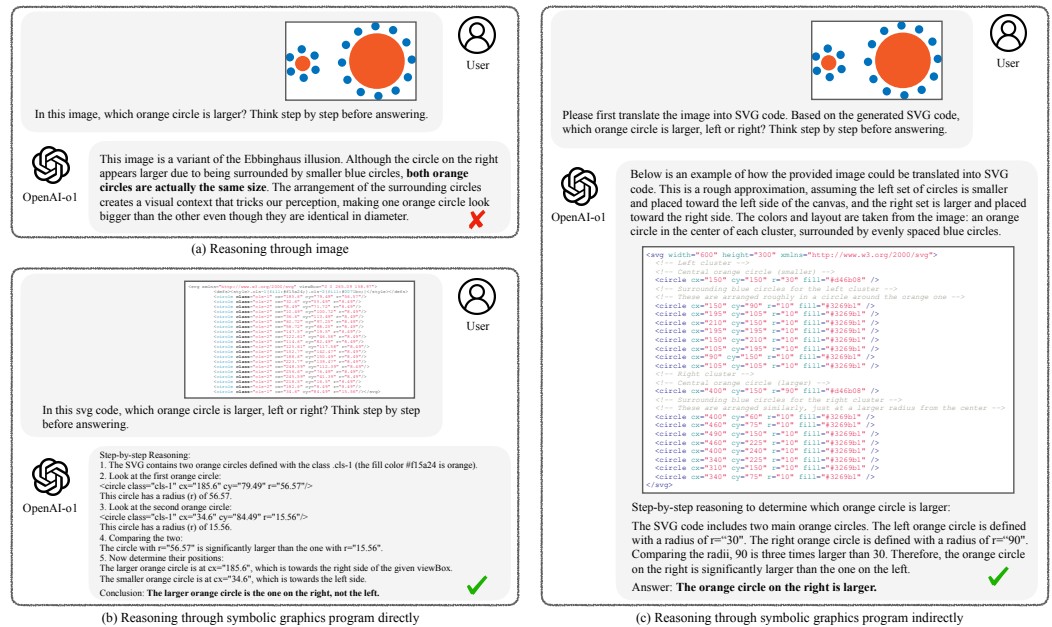

Figure 5: OpenAI-o1 still suffers from the spurious correlation from the Ebbinghaus illusion while reasoning over images (a). In contrast, OpenAI-o1 works perfectly fine while reasoning over symbolic graphics programs directly (b) or indirectly (c).

To highlight the importance of symbolic graphics programs, we give another example in Figure 5, where a powerful LLM like OpenAI-o1 can suffer from the visual spurious correlation, and in contrast, this spurious correlation can be avoided when using the symbolic graphics program as the visual representation. Specifically, we construct a visual example that resembles the Ebbinghaus illusion, but the conclusion differs from the classic Ebbinghaus illusion (*i.e.*, two orange circles are of the same size, although they look different) as we intentionally make one of the orange circle obviously larger than the other (shown in Figure 5). Then we feed this curated example to OpenAI-o1 and ask which orange circle is larger. We compare three cases: (a) image input; (b) symbolic program input; and (c) image input with *indirect symbolic program prompting*. Once we first ask the LLM to translate the image into symbolic program, the LLM will no longer suffer from the spurious correlation. The same

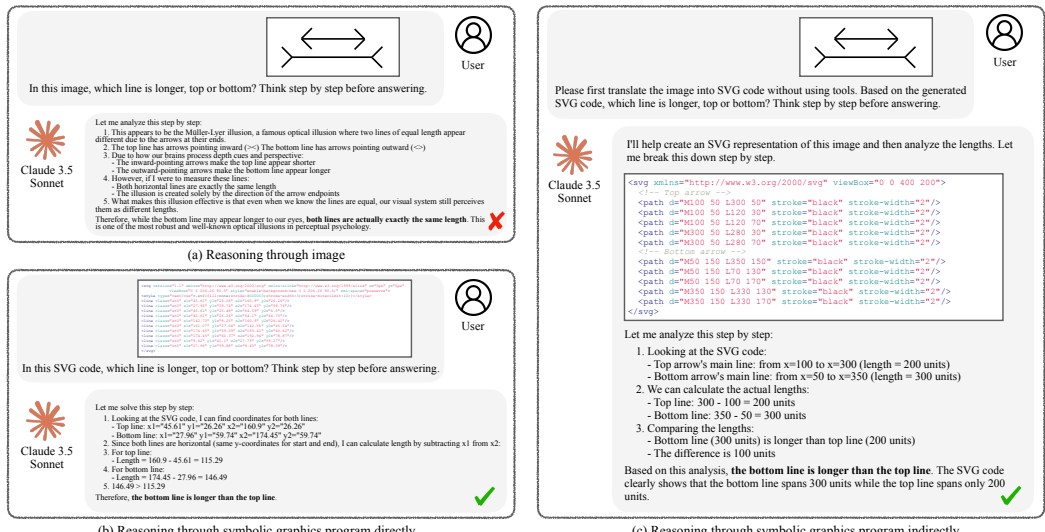

Figure 6: Claude-3.5-Sonnet also suffers from the spurious correlation from the Müller-Lyer illusion.

phenomenon also appears for the Müller-Lyer illusion using Claude-3.5-Sonnet (Figure 6). These examples validate the advantages of reasoning the visual world through symbolic graphics programs.

# 4    A BENCHMARK FOR SYMBOLIC GRAPHICS PROGRAM UNDERSTANDING

## 4.1    DATASET CREATION PIPELINE

To construct our benchmark, we need questions about a symbolic program, based on its rendered image. To build a large benchmark, it is essential to consider how we can scale up the question collection effectively, with minimal human effort. To this end, we use a powerful vision-language model (*e.g.*, GPT-4o) to generate semantic questions based on the rendered images, and then we inspect them manually to make sure that these questions are reasonable and the answer to them is correct. We also run a human study over a randomized set of 500 of

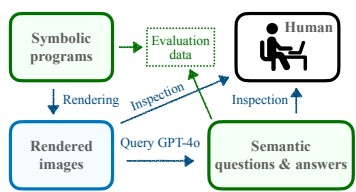

Figure 7: Dataset construction procedure.

the automatically generated questions along with the corresponding images, and find high agreement (see Appendix B). The overall procedure for our dataset creation is given in Figure 7. In this pipeline, the rendering of symbolic programs and the GPT-4o querying are both scalable and can be done with minimal human involvement. Human annotators then inspect the generated question-answer pairs based on the rendered image, which requires much less efforts than manually writing questions and answers. We emphasize that this program-question-answer triplet creation method is general, as it works for most of the symbolic graphics programs. SVG and 2D CAD programs can directly produce 2D images, so it is straightforward to use this pipeline. For 3D CAD programs, they produce 3D models and we first render them into 2D images with a few fixed camera positions. These rendered images are used to query GPT-4o, and the following procedures are identical to the SVG case.

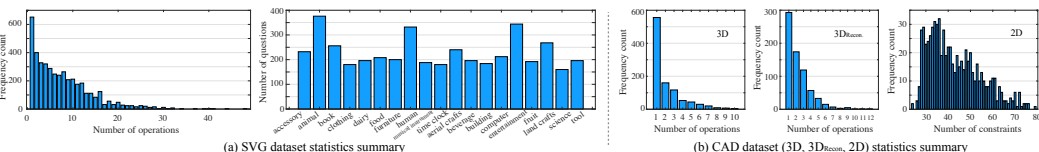

Figure 8: Key dataset statistics for both SVG and CAD programs. We show the distribution of the operation number in a program for both SVG and CAD data, together with the number of examples of each category in the SVG dataset.

## 4.2    BENCHMARKING SEMANTIC UNDERSTANDING

**SVG dataset statistics**. We collect $1,085$ SVG programs covering 19 categories, and each program has 4 semantic multiple-choice questions (with 4 options), resulting in a total of $4,340$ questions. We ensure that answers are evenly distributed across options. Dataset statistics are given in Figure 8(a). Our SVG benchmark consists of 5 types of questions, including "Semantic": $1,085$ questions, "Color": $864$ questions, "Shape": $1,217$ questions, "Count": $819$ questions, and "Reasoning":

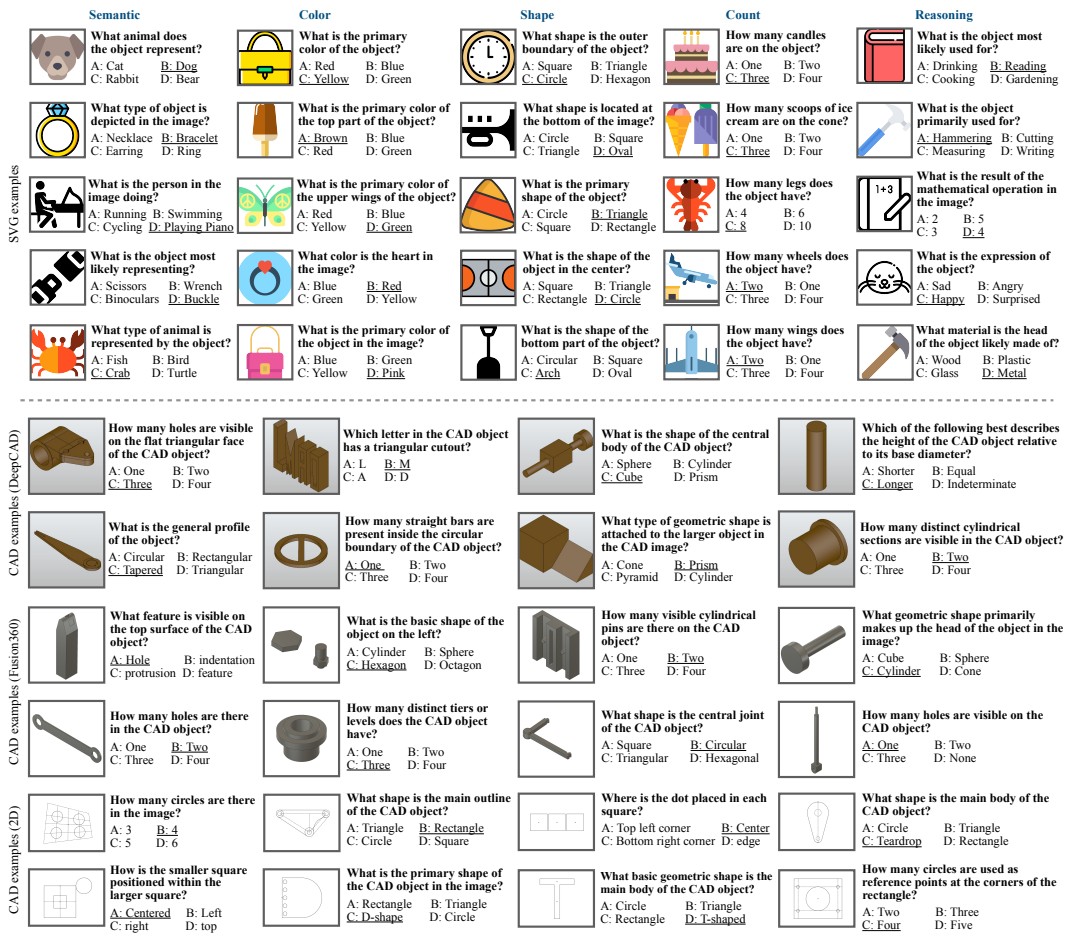

Figure 9: Example questions for SVG and CAD programs. Due to the space limit, we omit the programs and only show the rendered images.

355 questions. "Semantic" tests the global semantic meaning of the object represented by SVG codes, while the other four question types focus on detailed, local understanding of the object. "Color" is color-related questions about specific object parts, which evaluates the localization of the corresponding semantic part. "Count" is about counting the occurrences of certain patterns or semantic parts. "Shape" is about the shape of certain parts of the object, which is to find geometric shapes that closely resemble the object part. Figure 9 gives some SVG examples.

**CAD dataset statistics**. We collect 2, 400 CAD programs from three different datasets [101, 105, 86]. The CAD dataset consists of 1000 programs from DeepCAD [105], which forms the *3D* subset; 700 programs from the Fusion360 Reconstruction Dataset [101], which constitutes the *3D_{complex}* subset; and 700 programs from SketchGraphs [86], which makes up the *2D* subset (as shown in Table 1). Different from SVG, there is no generally established syntax for building CAD models from graphics codes; each of our 3 CAD subsets follows a different language syntax, with varying levels of complexity. When benchmarking LLMs for CAD tasks, we include domain-specific language syntax rules as part of the input prompt. The LLM is required to apply in-context learning of this syntax and answer the test questions. Then we feed the renderings to GPT-4o and generate one semantic multiple-choice question (with 4 options) and its answer. This gives us 2, 400 questions in total. We make sure that ground truth answers are evenly distributed across 4 options. Detailed dataset statistics are given in Figure 8(b). Some examples from our CAD dataset are provided in Figure 9.

**Experimental results and discussion**. We find that graphics program understanding, as we operationalize it here, is challenging. The average accuracy of all models (proprietary and open-source) is below 70% (ranging from 30% to 67%) on SVG and below 75% (ranging from 28% to 74%) on CAD. Among these, SVG makes it more difficult for the models to understand as these 2D graphics contain richer semantics. Significant performance improvements are observed in line with scaling laws [113], as larger model sizes consistently lead to gains across various open-source LLMs. For example, Llama-3's score is improved from 0.429 to 0.548 on SVG and from 0.633 to 0.694 when

its size increased from 8B to 70B, Qwen-1.5's from 0.376 to 0.499 on SVG and 0.486 to 0.632 on CAD with the size from 7B to 110B. We also notice consistent improvements given the same model size and model family but from different generations. For example, Qwen-72B from 0.466 to 0.537, Llama-8B from 0.429 to 0.465 and Llama-70B from 0.548 to 0.574. The consistent performance gain on both SVG and CAD indicates that semantic understanding of symbolic graphics programs is a fundamental capability that is aligned with the scaling law of LLMs.

Compared to the open-sourced LLMs that we consider here, proprietary models (GPTs) and (Claudes) outperform most of them by a large margin. Within the family of current most popular GPTs, we see a performance improvement with a 27% boost (from GPT-3.5's 0.498 to GPT-4's 0.633) when evaluating these GPT variants on our SGP-Bench. This result is aligned with the seeming improvement of reasoning ability in the GPT family, validating that SGP-Bench can well distinguish different LLMs. The overall best-performing model in both SVG and CAD is Claude 3.5 Sonnet.

The semantic understanding of graphics programs can be probed across different aspects, ranging from attribute-level investigations of "color" and "shape" to higher-level discussions of "semantics", "counting" and "reasoning". Our benchmark is designed to cover these investigations for LLMs. Most LLMs perform well on color-related questions, followed by shape-related questions, with "count" and "semantic" questions showing progressively lower performance. This consistency is intriguing, as it resembles the coarse-to-fine structure of visual information processing. "Color" is the most visually salient feature, "shape" understanding requires a finer grasp of global and local structures, and "count" and "semantic" questions demand deeper comprehension and knowledge. The difficulty curve is evident, with most open-source models achieving roughly half the accuracy on semantic questions compared to color questions. For instance, the best-performing open-source model, Llama3.1-405B, achieves 37.6% accuracy on semantics and 81.6% accuracy on color grounding tasks. While open-source models struggle with "semantic" questions, ChatGPT performs quite well, with semantics being their second-best category after color grounding.

## 4.3 BENCHMARKING SEMANTIC CONSISTENCY

LLMs are exposed to vast amounts of online SVG data. To investigate whether their semantic understanding ability is due to potential data leakage, we propose a semantic consistency test by introducing global translations or rotations to SVG graphics, ensuring SE(2) invariance. Such spatial interventions greatly alter the code representation, as SVG graphics consist of lines and Bezier curves with anchor points, and SE(2) operations change all numerical values in the code. However, the SVG's semantics—such as shape or color—remain unaffected by this perturbation. This allows us to examine how LLMs behave when the same vector graphics are presented with drastic code-numerical changes (see Appendix A.1). We perform

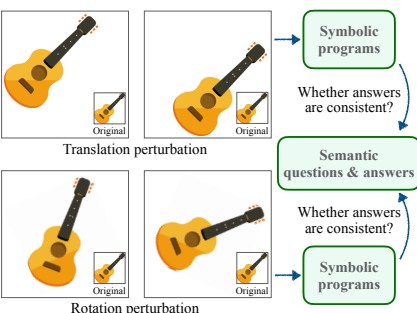

Figure 10: The semantic consistency test assesses if semantic understanding remains the same when the program is perturbed without semantically changing its rendered content. Image perturbations result in significant code-level changes, as symbolic programs use absolute coordinates.

non-trivial coordinate-level perturbations to the code, rather than using SVG transformation functions, to prevent shortcut learning by LLMs. Due to the nested structure of the tested SVG code, we visually inspect the perturbed renderings to ensure that the semantics remain unchanged after perturbation. If the model performs consistently under these perturbations, it suggests that its semantic understanding stems from a fundamental level of comprehension rather than trivial memorization.

**Dataset specifics**. We use our SVG dataset to evaluate the semantic consistency with respect to translation and rotation. For each SVG sample, we randomly choose 5 different translations (T) and rotations plus translations (SE(2), harder case), resulting in a visually small amount of spatial shifts of the rendered object, meaning nearly no changes in semantics, but **complete** change in SVG code numeric given the shift of SVG's anchor points and curves. Then we evaluate all the LLMs with the same question set of the SVG-Understanding benchmark but with these perturbed code inputs.

**Evaluation**. We measure the semantic consistency with two metrics: 1) the average accuracy of all perturbed SVG inputs' question-answering accuracy, showing the overall accuracy once the samples are intervened; and 2) the proposed "consistency score" that counts the average frequency of the most selected answer to each question for all groups of perturbed samples (where they were translated or rotated from the same SVG program). This score indicates how much the LLMs being consistent (no

| Model | SVG - Understanding | | | | | | SVG - Invariance | | | | CAD | | | |
|---|---|---|---|---|---|---|---|---|---|---|---|---|---|---|
| | Avg | Semantics | Count | Color | Shape | Reason | T Avg. | SE(2) Avg. | T Cons. | SE(2) Cons. | Avg | 3D | $3D_{complex}$ | 2D |
| *Open-source generic LLM* | | | | | | | | | | | | | | |
| Gemma-1.1-2B | 0.317 | 0.321 | 0.333 | 0.25 | 0.356 | 0.287 | 0.312 | 0.270 | 0.954 | 0.920 | 0.278 | 0.294 | 0.253 | 0.281 |
| Gemma-1.1-7B | 0.393 | 0.347 | 0.275 | 0.453 | 0.523 | 0.299 | 0.403 | 0.390 | 0.917 | 0.894 | 0.476 | 0.497 | 0.464 | 0.460 |
| InternLM2-7B | 0.382 | 0.279 | 0.324 | 0.570 | 0.431 | 0.299 | 0.381 | 0.381 | 0.788 | 0.772 | 0.480 | 0.551 | 0.446 | 0.411 |
| InternLM2-20B | 0.424 | 0.255 | 0.379 | 0.623 | 0.483 | 0.276 | 0.426 | 0.407 | 0.777 | 0.727 | 0.525 | 0.586 | 0.490 | 0.474 |
| InternLM2.5-7B | 0.421 | 0.273 | 0.317 | 0.598 | 0.515 | 0.282 | 0.419 | 0.404 | 0.809 | 0.819 | 0.562 | 0.639 | 0.506 | 0.509 |
| Yi-1.5-9B | 0.355 | 0.309 | 0.404 | 0.493 | 0.297 | 0.301 | 0.372 | 0.374 | 0.947 | **0.947** | 0.469 | 0.581 | 0.416 | 0.361 |
| Yi-1.5-34B | 0.443 | 0.308 | 0.364 | 0.644 | 0.523 | 0.234 | 0.446 | 0.423 | 0.845 | 0.819 | 0.583 | 0.649 | 0.563 | 0.510 |
| Aya-23-8B | 0.290 | 0.244 | 0.255 | 0.343 | 0.326 | 0.259 | 0.290 | 0.273 | 0.942 | 0.896 | 0.428 | 0.508 | 0.384 | 0.359 |
| Aya-23-35B | 0.442 | 0.307 | 0.354 | 0.648 | 0.511 | 0.318 | 0.451 | 0.434 | 0.898 | 0.857 | 0.488 | 0.551 | 0.429 | 0.457 |
| Command R-35B | 0.461 | 0.311 | 0.442 | 0.676 | 0.495 | 0.341 | 0.478 | 0.443 | 0.833 | 0.803 | 0.536 | 0.579 | 0.509 | 0.504 |
| Command R-104B | 0.500 | 0.339 | 0.449 | 0.727 | 0.565 | 0.341 | 0.521 | 0.477 | 0.917 | 0.875 | 0.583 | 0.634 | 0.570 | 0.524 |
| Qwen-1.5-7B | 0.376 | 0.226 | 0.317 | 0.563 | 0.471 | 0.234 | 0.371 | 0.382 | 0.792 | 0.780 | 0.486 | 0.560 | 0.426 | 0.443 |
| Qwen-1.5-32B | 0.494 | 0.307 | 0.501 | 0.713 | 0.552 | 0.310 | 0.512 | 0.492 | **0.972** | 0.938 | 0.575 | 0.664 | 0.567 | 0.456 |
| Qwen-1.5-72B | 0.466 | 0.299 | 0.319 | 0.698 | 0.598 | 0.265 | 0.474 | 0.461 | 0.883 | 0.854 | 0.600 | 0.658 | 0.590 | 0.526 |
| Qwen-1.5-110B | 0.499 | 0.324 | 0.431 | 0.734 | 0.560 | 0.332 | 0.486 | 0.470 | 0.839 | 0.821 | 0.632 | 0.711 | 0.607 | 0.546 |
| Qwen-2-72B | 0.537 | 0.373 | 0.426 | 0.770 | 0.630 | 0.372 | 0.520 | 0.491 | 0.869 | 0.852 | 0.692 | 0.753 | 0.669 | 0.630 |
| Mistral-7B v0.3 | 0.417 | 0.304 | 0.324 | 0.624 | 0.470 | 0.296 | 0.434 | 0.417 | 0.919 | 0.895 | 0.495 | 0.551 | 0.481 | 0.429 |
| Mistral-NeMo-12B | 0.449 | 0.296 | 0.355 | 0.652 | 0.548 | 0.296 | 0.480 | 0.443 | 0.894 | 0.855 | 0.568 | 0.623 | 0.549 | 0.510 |
| Mistral-Large2-123B | 0.572 | 0.389 | 0.558 | 0.814 | 0.635 | 0.408 | 0.577 | 0.540 | 0.889 | 0.847 | 0.710 | 0.755 | 0.716 | 0.640 |
| LLama3-8B | 0.429 | 0.304 | 0.372 | 0.626 | 0.484 | 0.293 | 0.426 | 0.410 | 0.905 | 0.873 | 0.550 | 0.633 | 0.472 | 0.512 |
| LLama3-70B | 0.548 | 0.364 | 0.496 | 0.749 | 0.645 | 0.369 | 0.559 | 0.525 | 0.905 | 0.874 | 0.634 | 0.694 | 0.619 | 0.566 |
| LLama3.1-8B | 0.465 | 0.339 | 0.385 | 0.667 | 0.533 | 0.268 | 0.464 | 0.448 | 0.821 | 0.806 | 0.574 | 0.626 | 0.539 | 0.534 |
| LLama3.1-70B | 0.574 | 0.400 | 0.543 | 0.788 | 0.659 | 0.411 | 0.584 | 0.554 | 0.856 | 0.825 | 0.688 | 0.739 | 0.663 | 0.641 |
| LLama3.1-405B | 0.580 | 0.376 | **0.584** | 0.816 | 0.647 | 0.389 | 0.570 | 0.548 | 0.840 | 0.817 | 0.717 | 0.767 | 0.700 | 0.661 |
| *Open-source code LLM* | | | | | | | | | | | | | | |
| CodeQwen1.5-7B | 0.301 | 0.245 | 0.262 | 0.344 | 0.387 | 0.245 | 0.327 | 0.324 | 0.883 | 0.859 | 0.376 | 0.419 | 0.350 | 0.340 |
| DeepSeek-Coder-V2-16B | 0.451 | 0.309 | 0.379 | 0.637 | 0.548 | 0.268 | 0.496 | 0.476 | 0.902 | 0.867 | 0.547 | 0.611 | 0.521 | 0.481 |
| Codestral-22B-v0.1 | 0.491 | 0.309 | 0.446 | 0.698 | 0.581 | 0.321 | 0.503 | 0.470 | 0.860 | 0.816 | 0.602 | 0.659 | 0.577 | 0.547 |
| *Proprietary models* | | | | | | | | | | | | | | |
| GPT-3.5 Turbo | 0.498 | 0.319 | 0.451 | 0.729 | 0.577 | 0.338 | 0.509 | 0.492 | 0.897 | 0.870 | 0.576 | 0.654 | 0.530 | 0.510 |
| GPT-4 Turbo | 0.609 | 0.764 | 0.539 | 0.832 | 0.687 | 0.412 | 0.606 | 0.576 | 0.867 | 0.835 | 0.716 | 0.762 | 0.694 | 0.674 |
| GPT-4o mini | 0.585 | 0.398 | 0.504 | 0.791 | 0.709 | 0.414 | 0.595 | 0.561 | 0.881 | 0.852 | 0.659 | 0.737 | 0.612 | 0.594 |
| GPT-4o | 0.633 | **0.787** | 0.553 | 0.832 | 0.696 | 0.471 | 0.625 | 0.586 | 0.878 | 0.844 | 0.733 | 0.782 | 0.711 | 0.686 |
| Claude 3 Haiku | 0.486 | 0.264 | 0.398 | 0.750 | 0.610 | 0.301 | 0.496 | 0.476 | 0.902 | 0.867 | 0.612 | 0.677 | 0.581 | 0.549 |
| Claude 3 Sonnet | 0.565 | 0.375 | 0.503 | 0.803 | 0.657 | 0.395 | 0.582 | 0.566 | 0.875 | 0.838 | 0.644 | 0.673 | 0.649 | 0.599 |
| Claude 3.5 Sonnet | **0.674** | 0.505 | **0.584** | **0.891** | **0.758** | **0.527** | **0.670** | **0.649** | 0.903 | 0.870 | **0.742** | 0.769 | **0.727** | **0.717** |

Table 1: Performance of various LLMs on SGP-Bench. This table evaluates how well models understand SVG inputs ('SVG - Understanding') and their behavior under random perturbations of these inputs ('SVG - Invariance'). It also assesses 3D & 2D semantic understanding of CAD code. We found the results demonstrate the "scaling law", with larger LLMs in the same family showing superior performance. Bold texts indicates performance with **1st rank**, and underlined texts indicates performance with 2nd rank and 3rd rank.

matter right or wrong) regardless of the drastic program change. If the score is close to 1, it means all the predictions are the same even with totally different input codes.

**Experimental results and discussion**. Our experiments with the SVG-Invariance benchmark demonstrate that most LLMs exhibit robust semantic understanding of graphics programs under translation (T) and translation + rotation (SE(2)) perturbations. In Table 1, most of the LLMs achieve over 80% consistency in both perturbation settings, with half of the models exceeding 90% consistency. Not only do the models remain consistent in their predictions under perturbations, but their performance on perturbed inputs also shows minimal fluctuation compared to their performance on the SVG-Understanding benchmark. We posit that this indicates that the semantic understanding ability that we evaluate of LLMs is unlikely due to data leakage, but rather, could stem from a potential foundational capability to interpret the semantics of deterministic, symbolic graphics programs. Additionally, we assess the structural alterations introduced by our perturbation operation by calculating the tree edit distance between the original and perturbed code. Our findings indicate that the perturbation leads to varying levels of structural changes in the code. However, we observe no significant correlation between the degree of structural modification and the consistency performance (see Table 2).

| TED | T Cons. | SE(2) Cons. |
|---|---|---|
| 5-10 | 0.890 | 0.833 |
| 20-25 | 0.882 | 0.822 |

Table 2: Consistency with varying tree edit distance (TED) between original and modified codes.

## 4.4 PREDICTION ENTROPY OF LLMS AND HUMANS

To study the consensus of LLMs, we compute the average prediction entropy on 500 symbolic programs using GPT-4o, LLama3-8B, LLama3-70B, Mistral-7B, Yi-1.5-34B, Gemma-1.1-7B and Qwen-1.5-72B. We conduct a human experiment on the rendered

Figure 11: Comparison of prediction entropy.

images of these programs and collect the answers (each question has at least 5 participants, see Appendix B). Figure 11 shows that humans have strong consensus when answering questions based

on images, while LLMs show low consensus when answering questions based on symbolic programs. This implies that LLMs may have different inner working mechanisms to understand symbolic programs. We are excited by future work to better investigate this difference.

# 5    IMPROVING LLMS WITH SYMBOLIC INSTRUCTION TUNING

**Generating symbolic instruction data**. Inspired by how visual instruction tuning [56] enables large vision-language models to understand images with visual-question-answering (VQA) data, we design a new method to perform symbolic instruction tuning for LLMs to better bridge the gap between the semantic understanding and symbolic reasoning over the graphics programs. To our knowledge, there exist no semantic instruction datasets directly over symbolic graphics programs, After rendering these symbolic graphics programs into images, we can easily query powerful vision-language models (*e.g.*, GPT-4o is used in our case) to obtain a detailed semantic captioning based on the rendered image. The intuition is straightforward, as we want to build an intrinsic connection between semantic description and symbolic programs. The instruction data is created in a similar spirit to our benchmark. We leverage the correspondence between symbolic problems and graphics content, and then use the rendered images to obtain semantically rich description. Following this idea, we construct the first semantic description dataset for symbolic graphics programs. Specifically, for each image rendered from a symbolic graphics program, we prompt GPT-4o to produce a semantically-rich description. Finally, we collect a dataset that contains detailed semantic descriptions for 72K symbolic programs. Moreover, our SIT data can also be used in a reverse fashion (rev-SIT), *i.e.*, rephrasing the answer as the new question and the question as the new answer. Figure 12 shows the comparison between original and reverse SIT data.

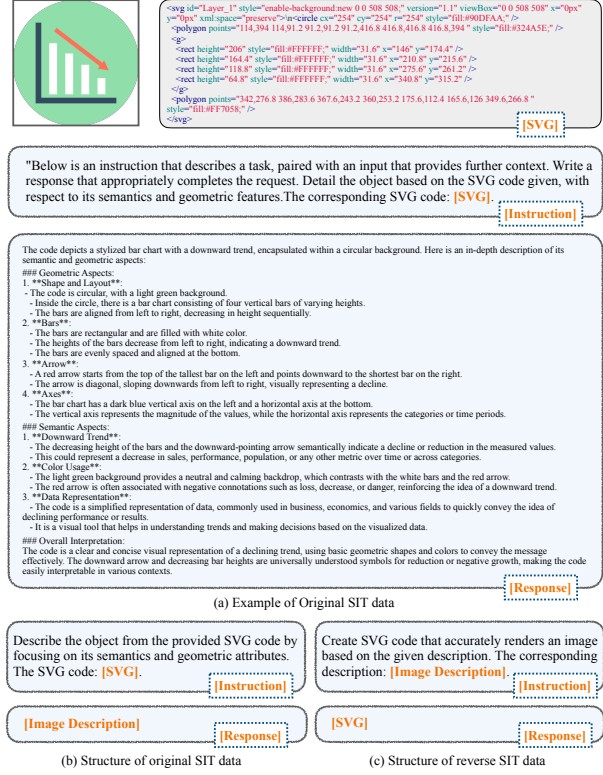

Figure 12: Comparison between original and reverse SIT data.

**Supervised finetuning with symbolic instruction data**. We generally follow the standard instruction fine-tuning procedure (including the default hyperparameter settings) from Alpaca [93] and use supervised finetuning to train open-source LLMs with our own symbolic instruction data. To facilitate future research, our symbolic instruction data is also made publicly available.

**Experimental results**. We perform supervised finetuning on Llama-3.1-8B with orthogonal finetuning [60, 75] to demonstrate the effectiveness of SIT. Here we use the original SIT data (no rev-SIT data is used). In Appendix E.2, we provide results on both Llama-3-8B and Gemma-7B to show that the performance gain is agnostic to the base LLM. The performance of LoRA [36] is also given in Appendix E.2 (only slightly worse than OFT). From the experimental results in Table 3, we observe that SIT has consistently improved

| Dataset Size | Accuracy |
|---|---|
| Original | 46.5 |
| SIT-10k | 48.0 (+1.3) |
| SIT-25k | 50.3 (+3.6) |
| SIT-40k | 51.2 (+4.5) |
| SIT-55k | **51.4** (+4.7) |

Table 3: Performance of SIT.

the semantic graphics program understanding of LLMs, increasing the performance of Llama-3.1-8B from 46.7% to 51.4% with 55K instruction question-answer pairs. With more instruction data (from 10K to 55K), the performance also increases. We note that Llama-3.1-8B achieves competitive performance among all open-source LLMs after being finetuned with SIT. The performance is already better than GPT-3.5t. The finetuning results demonstrates that the ability of symbolic graphics program understanding can be improved with SIT. However, the improved performance of Llama-3.1-8B remains worse than Llama-3.1-70B, indicating that the tested ability is fundamental and differences between models of varying scales cannot be easily leveled by finetuning on benchmark-like data.

## 6 SIT CAN IMPROVE GENERAL REASONING ABILITY

Since Figure 4 shows that the ability to understand symbolic graphics programs is associated with some fundamental reasoning abilities of LLMs, we are interested in whether symbolic graphics programs can be used as a novel data source for building better instruction tuning datasets, which can help to improve the general reasoning ability of LLMs. To verify this, we test the instruction-tuned models on a variety of popular LLM benchmarks, including benchmarks focusing on natural language understanding (XNLI [17], IFEval [119], HellaSwag [117], C-Eval [39], CoQA [80], MMLU [34], SQuAD2.0 [77]), generic reasoning (BigBenchHard [92], PIQA [7], AGIEval [118]) and mathematical reasoning (Arithmetic [9], MathQA [3], GSM8k [14], ASDiv [71]).

**Experimental results and discussion**. We use the Llama-3.1-8B model (without instruction tuning), and the baseline is finetuned with Open-Instruct (OI) [97] that contains 143K question-answer pairs (details in Appendix E.1). We evaluate whether finetuning with SIT data can improve general reasoning by testing three ways of using SIT data: (1) mixing original SIT data into OI; (2) mixing the reverse SIT data into OI; (3) mixing both original and reverse SIT data into OI. The results are given in Table 4. We can observe that mixing SIT data can generally improve the instruction following and the reverse usage of SIT data (*i.e.*, symbolic graphics program generation) can improve a set of reasoning abilities that are complementary to symbolic graphics program understanding. The mixture of both original and reverse SIT data often achieves

| Benchmark | OI | OI-SIT | OI-rev-SIT | OI-mixed-SIT |
|---|---|---|---|---|
| XNLI | 41.8 | **43.3** (+1.5) | 43.1 (+1.3) | 42.9 (+1.1) |
| IFEval$_{prompt}$ | 14.8 | 16.3 (+1.5) | **18.3** (+3.5) | 16.6 (+1.8) |
| IFEval$_{inst.}$ | 24.9 | 28.9 (+4.0) | **30.5** (+5.6) | 29.6 (+4.7) |
| HellaSwag | 60.0 | 60.2 (+0.2) | **60.5** (+0.5) | 60.4 (+0.4) |
| C-Eval | 46.4 | 47.9 (+1.5) | 48.0 (+1.6) | **48.1** (+1.7) |
| MMLU | 60.4 | 61.0 (+0.6) | 61.1 (+0.7) | **61.6** (+1.2) |
| SQuAD2.0 | 28.9 | 28.7 (-0.2) | **31.6** (+2.7) | 29.9 (+1.0) |
| BBH | 59.5 | 60.7 (+1.2) | 60.2 (+0.7) | **61.2** (+1.7) |
| PIQA | 79.9 | 80.3 (+0.4) | 80.3 (+0.4) | **80.4** (+0.5) |
| AGIEval | 23.7 | 30.3 (+6.6) | **31.6** (+7.9) | 29.2 (+5.5) |
| Arithmetic | 89.8 | **91.8** (+2.0) | 90.1 (+0.3) | **91.8** (+2.0) |
| MathQA | 39.3 | 40.4 (+1.1) | 40.3 (+1.0) | **40.7** (+1.4) |
| GSM8k | 48.2 | 50.7 (+2.5) | 51.0 (+2.8) | **51.5** (+3.3) |
| CoQA | 67.9 | **69.1** (+1.2) | 68.7 (+0.8) | **69.1** (+1.2) |
| ASDiv | 18.5 | **21.8** (+3.3) | 20.1 (+1.6) | 21.3 (+2.8) |

Table 4: Results on a variety of popular LLM evaluation benchmarks when performing instruction tuning with or without SIT. The Open-Instruct (OI) dataset serves as our baseline.

better performance than the OI baseline, the OI + SIT baseline and the OI + rev-SIT baseline. These results are consistent with recent findings that training on code can enhance reasoning ability [66, 4] and mathematical understanding [88]. Symbolic graphics programs, a specialized form of code, are used to generate visual graphics content. Like traditional code, they possess a hierarchical structure, but unlike typical programs that produce numerical outputs, symbolic graphics programs generate outputs rich in semantic information, encompassing multiple challenging reasoning tasks such as component localization, color identification, affordance prediction, and semantic and geometric understanding. For instance, answering a question like "What is the object primarily used for?" requires LLMs to first semantically identify the object and then determine its usage. This process involves multiple interconnected reasoning steps, where an error in any one of them leads to an incorrect final answer. SIT enhances reasoning abilities by interpreting low-level graphics programs through high-level natural language descriptions. From Figure 12(a), we see that symbolic graphics program descriptions are highly detailed and semantic—qualities often lacking in general programs.

## 7 A CRITICAL VIEW ON CURRENT LLM'S CAPABILITY

| Method | Accuracy |
|---|---|
| LLama3-70B | 10.0 |
| Qwen-1.5-110b | 10.0 |
| Qwen-2-70b | 11.3 |
| GPT-3.5t | 10.2 |
| GPT-4t | 10.6 |
| GPT-4o | **13.0** |

Table 5: Accuracy of LLMs on SGP-MNIST.

Despite the observed remarkable capability of LLMs to perform complex, multi-step reasoning over symbolic programs, it is evident that there remains substantial potential for further advancements. We provide an intriguing experiment to demonstrate that SVG programs can be quite difficult for LLMs to understand such that even if the corresponding rendered images are fairly easy for humans to recognize, all these powerful LLMs still fail dramatically.

Specifically, we construct symbolic graphics programs that can produce MNIST-like images, as shown in Figure 4 (and Appendix A.1). Our SGP-MNIST dataset contains 1,000 symbolic graphics programs (100 per digit), each asking which digit the SVG program represents. The results are given in Table 5. Even the powerful GPT-4o can only achieve an accuracy slightly higher than the chance-level. The MNIST-like symbolic program presents a unique challenge due to the absence of semantic components for LLMs to reason upon. Instead, it comprises convoluted, irregular path trajectories that resemble handwritten digits. Additionally, the program contains not only single paths but enclosed loops to represent the "thickness" of digits, demanding precise path planning by the LLMs. For instance, the digit 1 is not represented as a single "line" but rather as an elongated loop, which must be distinguished from more oval-shaped loops, such as those representing the digit 0.

Without prior knowledge of digit "thickness," the LLM must infer this distinction through detailed reasoning over the loop structures, further elevating the complexity of the task. The chance-level performance suggests that how LLMs understand SVG programs is very different from how humans understand images; better understanding similarities and differences in human and machine reasoning is important if we are to build systems that can appropriately work with us [16]. There are many exciting yet totally unexplored problems in this task, and our benchmark can serve as a stepping stone to improving symbolic graphics program understanding for LLMs.

# 8 RELATED WORK AND ACKNOWLEDGMENT

**Symbolic graphics programs**. Generating visual data by procedural modeling with symbolic programs has been essential to computer graphics since its inception, particularly for 2D shapes and 3D geometry. See [83] for an overview. Common program types include constructive-solid geometry (CSG) [21, 45, 81, 89, 112], computer-aided design (CAD) [30, 50, 51, 87, 108], vector graphics (*e.g.*, SVG) [78, 79], L-systems [33], and customized domains [22, 95, 18, 37, 24, 69]. Among these, SVGs are constructed from primitive shapes like vector paths, curves, or polygons. Central to SVGs is the vector path, providing detailed control over graphics and geometry primitives. Similarly, procedural 3D geometric modeling, particularly in CAD applications, involves parameterized operations to produce geometry. Datasets like the ABC [47] and Fusion 360 Gallery [102] offer hierarchical decomposition, joints, contact surfaces, construction sequences, and shape segmentation based on modeling operations. Our paper focuses on graphics programs of SVG and CAD by introducing a new semantic understanding task that requires a challenging reasoning over the programs.

**Graphics program understanding and generation**. As graphics programs often provide compact, scalable and potentially more semantic descriptions compared to raw pixels and voxels, it has been widely explored to discover graphics programs for 2D images like 2D hand drawings and synthetic patterns [98, 99, 89, 23, 82, 29, 87], for 3D objects represented in voxels and meshes [29, 43, 95, 8, 102, 89, 23] and for 3D scenes represented by multi-view images [68, 49, 104, 111, 69, 61, 53, 31, 30, 46, 20]. [104] infers custom-designed markup code from images that can be easily translated to renderer-friendly inputs. In follow-up work, [111] explore how graphics programs can be used for visual question answering (VQA). Recently, [48] has advanced this direction by examining large language models (LLMs) for synthesizing graphics programs to reconstruct visual input. In contrast, we benchmark LLMs to perform semantic-level question answering, similar to VQA, but use graphics programs as input without relying on any visual modality.

**Large language models**. LLMs have demonstrated growing potential in many applications, ranging from mathematical problem solving and theorem proving assistance [65, 114, 116, 15] to aiding biological discovery [67, 27, 94, 59]. Applying LLMs for programming tasks is also a popular direction of research. Specifically, many works have explored topics such as code retrieval [26], automated testing [19, 62, 109], repairing [44, 106, 41, 42, 100], documentation [13, 2], and generation [11, 5, 54, 28, 72]. These abilities of understanding and generating programs are usually gained from pretraining or finetuning on large datasets of code. Our work investigates LLMs' capability of understanding symbolic graphics programs, which differs significantly from the prior works since the semantic meaning of graphics programs are often defined visually by their corresponding graphics.

**Relevant benchmarks and datasets**. Many benchmarks have evaluated different aspects of LLMs: AI safety/ethics [107, 38], out-of-distribution performance [115, 110], API/tool usage [52], code generation [35], *etc*. Perhaps the most relevant aspect of LLM evaluation to our task is (non-graphics) program understanding abilities [96, 44, 35, 64, 57, 63, 90, 34, 11]. As graphics programs can be rendered into images, it is also highly relevant to investigate how vision-language models are capable of visual understanding [12, 1, 74, 40, 70, 32, 6, 91, 85, 120]. For SVG programs, [10] studies whether LLMs can understand them and [122] introduces a concurrent benchmark for this purpose. Different from existing benchmarks, SGP-Bench is one of the first benchmarks to evaluate the semantic understanding of general graphics programs.

**Acknowledgement**. The authors would like to thank Yao Feng and Yandong Wen for helpful suggestions. Additionally, HF would like to thank Hanqi Zhou for her support throughout the project, during which time they became engaged (Fun fact: HF and Hanqi Zhou got married a few hours before the submission deadline). The diamond ring featured in Figure 1 symbolizes this joyous personal milestone and is courtesy of the entire team. This work was supported in part by the German Federal Ministry of Education and Research (BMBF): Tubingen AI Center, FKZ: 01IS18039B, and

by the Machine Learning Cluster of Excellence, EXC number 2064/1 – Project number 390727645. WL was supported by the German Research Foundation (DFG): SFB 1233, Robust Vision: Inference Principles and Neural Mechanisms, TP XX, project number: 276693517. KMC acknowledges support from the Marshall Scholarship and Cambridge Trust. AW acknowledges support from a Turing AI Fellowship under grant EP/V025279/1, The Alan Turing Institute, and the Leverhulme Trust via CFI. MJB has received research gift funds from Adobe, Intel, Nvidia, Meta/Facebook, and Amazon. MJB has financial interests in Amazon, Datagen Technologies, and Meshcapade GmbH. While MJB is a consultant for Meshcapade, his research in this project was performed solely at, and funded solely by, the Max Planck Society.

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

# Appendix

## Table of Contents

# A BENCHMARK DETAILS

We adopt implementations from other projects to build our SGP-Bench. We follow the implemntation of *MathVista*[1] for querying GPT or open-sourced Llama3.1-8B and perform LLM-based answer extraction, *vLLM*[2] for efficient model inference and *simple-evals*[3] for a unified benchmarking framework.

Our data license follows the license of the original license of our data source.

## A.1 DATA PREPARATION

**SGP-Bench (SVG).** Our SVG data are sampled from kaggle *SVG Icons* dataset[4] and we build our SGP-Bench (SVG) using text prompts from F.1. The original data from kaggle *SVG Icons* is crawled from *SVGrepo*[5]. The website is merely an aggregator, so it follows that any content on it must at least be licensed permissively enough for SVGrepo to distribute it, and therefore it is acceptable to distribute as part of a collection in our benchmark. Refer to the SVGrepo license for further details.

**SVG Invariance:** We use beautifulsoup and SvgLib to process the SVG XML code to perform translation and rotation perturbations for the invariance investigation, a visual sample can be found in Fig. 17. Specifically, as we assume that all XML elements in each SVG figure do not possess any "transform" attribute as it complicates the augmentation process. For elements that can be fully specified by coordinates (*e.g.*, <rect>, <polygon>), we perform augmentation by perturbing these coordinates. For <path> elements in which path information is fully specified in "d" attributes, we first turn all relative operations (*e.g.*, "l 2 3", meaning that draw a line from the current position $(x, y)$ to the position $(x + 2, y + 3)$) into absolute ones, and later perturb the coordinates but not other path attributes. As mentioned in the main paper, small spatial perturbations can drastically change the numerics of the SVG XML code (See Section C for more details).

**SGP-Bench (CAD).** Our CAD (3D) sequences data are sampled from *DeepCAD* [105] datasets, which contains around 180k manually constructed CAD sequences that are originally from the ABC dataset [47]. We manually sample 1000 sequences and use pythonocc (OpenCASCADE) to verify and normalize these CAD sequences, then render the front and back view of the 3D CAD models. Since all the CAD models are from the OnShape platform, the copyright of the CAD models is owned by their creators. For licensing details, see Onshape Terms of Use 1.g.ii. Our CAD (3D$_{complex}$) sequences are sampled from *Fusion360 Reconstruction Dataset* [101] datasets, which contains 8,625 sequences, with more complex curve operations for constructing sketches. Our CAD (2D) sequences are sampled from *SketchGraphs* [86] dataset, which consists of 15 million sketches extracted from real-world CAD models. The major difference is it consists of 2D CAD sketches without Extrusion operations.

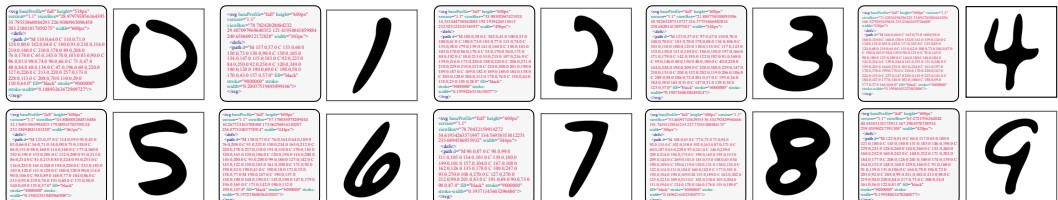

Figure 13: Examples of our SGP-MNIST challenge, hand-written digit constructed by SVG programs.

**SGP-MNIST.** The MNIST SVG data is sampled from the kaggle *MNIST-SVG* dataset[6]. We randomly sample 1000 samples from *MNIST-SVG* (100 samples per digit category) to build our SPG-MNIST benchmark. The data comes with CC BY-SA 3.0 license.

---

[1] https://github.com/lupantech/MathVista

[2] https://github.com/vllm-project/vllm

[3] https://github.com/openai/simple-evals

[4] https://www.kaggle.com/datasets/victorcondino/svgicons

[5] https://www.svgrepo.com/page/licensing/

[6] https://www.kaggle.com/datasets/jacekpardyak/mnist-svg

## A.2 Evaluation protocol

**Inference.** To conduct massive-scale evaluation (8000+ question samples with 16 models), we leverage vLLM[7] to perform high-throughput and memory-efficient inference for all the open-source LLMs. And we use OpenAI API for evaluating different variants of the GPT models. We deploy the vLLM inference engine as a server that also uses the same format as OpenAI API to achieve a unified testing framework for both GPT and all open-source models. The vLLM inference engine is deployed on a node with 8 NVIDIA H100 80G GPUs.

**Evaluation.** We benchmark the performance of all the models via the question answering accuracy. Following the common protocol [34], we ask the model to generate the answer sentence in a formatted way (see text prompt examples in F.2). Then, we extract the target answer from the output sentence by parsing the answer according to its position. If the extracted answer matches the ground truth, it will count as 1, otherwise it's 0. We use the average accuracies for the results in Table 1.

**Enhanced answer extraction with LLM:** In our experiment, we found the Symbolic Instruction Tuning makes the model less capable in following the formatting instruction. This is likely due to our fine-tuning only uses symbolic graphics description, which causes the model to forget its instruction following skill. Therefore, the model after SIT often answers questions correctly but in a different format. This affects the aforementioned format-based answer extraction. For example, given a color-grounding question of the input subject, the formatted answer should be "The answer is A) Yellow.", however, the model outputs "The car is yellow". To mitigate this issue, we follow the GPT-enhanced answer extraction of Mathvista[8], where we present both the question options and model's output to GPT4 to extract the answer in the formatted way. A 5-shot CoT is also applied here to augment the robustness of extraction process (More details in F.2). The results on Table 3 are obtained with the enhanced answer extraction. More details of the SIT evaluation can be found in Appendix E.

## A.3 Evaluated model specs

Here we list the model details of all LLMs we evaluated in the SGP-Bench, their performance are demonstrated in the table 3. Generally, we evaluated 3 types of LLMs, the representative open-sourced LLMs from tech giants and start-ups, the code-specific LLMs that were built for code generation and understanding and the strongest proprietary models from the GPT and Claude family.

### A.3.1 Open-sourced LLMs

**Gemma-1.1-2B/7B** Gemma is a suite of lightweight, advanced open models created by Google DeepMind and other teams across Google, it's the best performing model at it class when it's released on Feb 21, 2024. Primarily designed for text generation, Gemma models come in multiple sizes, i.e. 2B / 7B, to fit various computing resources and deployment needs. The models are trained on 3T (2B) / 6T (7B) tokens of primarily-English data from web, mathematics, and code. It is based on a transformer decoder with context length of 8192 tokens. It leverages Multi-Query Attention, RoPE Embeddings, GeGLU Activations and RMSNorm. The Gemma models are using architectures. data and training recipes inspired by the Gemini model family. The models are available in two versions: a pretrained version and an Instruction-tuned version, the latter refined through human language interactions to perform well in conversational roles, similar to a chat bot. We only test and perform the symbolic instruction tuning on the Instruction tuned version.

**Mistral-0.3-7B** The Mistral-0.1-7B from Mistral AI is released September 27, 2023 and marked as the best 7B model at the time. The 0.1 version model features a 8k context window, with Grouped-query attention (GQA) for faster inference and SWA for handling longer sequences more effectively at a reduced computational cost. The model is further updated to 0.3 version in May 21, 2024, upgrading its context length to 32k, its vocabulery size and RoPE theta, but the SWA is removed in this version.

**Mistral-NeMo and Mistral-Large2** Mistral NeMo is a 12B large language model built by Mistral AI with a context window size of up to 128k tokens. Mistral NeMo is trained with quantization

---

[7]https://github.com/vllm-project/vllm
[8]https://github.com/lupantech/MathVista

awareness, allowing FP8 inference without any loss in performance. Mistral NeMo uses a new tokenizer, Tekken, based on Tiktoken, which enables more efficient compression of natural language text and source code, compared with previous Mistral model series.

Mistral Large 2 is the new generation of Mistral AI's flagship model, with a model size of 123 billion parameters. Especially, Mistral Large 2 is trained on a very large proportion of code data, resulting in state-of-the-art performance, on par with proprietary models like GPT-4o or Clause Opus.

**Yi-1.5-9B/34B** The Yi model family, developed by LLM-focused startup 01.AI, includes 6B and 34B pretrained language models. Their performance is attributed to high-quality data from meticulous data-engineering efforts. For pretraining, 3.1 trillion tokens of English and Chinese corpora were constructed using a cascaded data deduplication and quality filtering pipeline. Finetuning involved a carefully refined instruction dataset of fewer than 10K instances, each verified by dedicated machine learning engineers. Built on Transformer architecture, Yi models feature Grouped-Query Attention (GQA), SwiGLU activation, and RoPE with an adjusted base frequency (RoPE ABF). The Yi-6B base model, with 32 layers, was scaled up to the Yi-9B model, which has 48 layers, by duplicating the original 16 middle layers (layers 12-28). We hence test the Yi-9B model together with the 34B version in SGP-Bench.

**InternLM2-7B/InternLM2-20B/InternLM2.5-7B** InternLM2 is a open-sourced LLM model series developed by Shanghai AI laboratory, with a context window length of 200K.

InternLM2.5 is an open-sourced, 7 billion parameter base and chat model with a context window size of 1M. It supports gathering information from more than 100 web pages and has in general a very strong capability in tool utilization.

**Aya-23-8B/35B** Aya 23 is an open-source LLM model series developed by C4AI, featuring advanced multilingual capabilities. Aya-23 is fine-tuned (IFT) to follow human instructions and supports a context window length of 8192 tokens.

**Command R-35B/104B** C4AI Command-R is a research release of a 35B large language model with open weights, optimized for a variety of use cases including reasoning, summarization, and question answering. Command-R has the capability for multilingual generation evaluated in 10 languages and highly performant RAG capabilities. It supports a context length of 128K.

C4AI Command-R+ is an open-source multilingual LLM with enhanced features, including Retrieval Augmented Generation (RAG) and tool usage for automating complex tasks. Command-R+ excels in multi-step tool usage, allowing the model to combine various tools across multiple steps to complete sophisticated tasks.

**Qwen-1.5-7B/32B/72B/110B** Qwen, released in April 2024, developed by Alibaba Cloud, is a series of transformer-based large language models pre-trained on diverse data, including web texts, books, code, and more, over 2.2 trillion tokens. The Qwen1.5 series includes various sizes of decoder models, each available as a base and aligned chat model, supporting long context lengths (8K tokens for 1.8B, 7B, and 14B models, and 32K tokens for the 72B model). It outperforms similar-scale open-source models on various Chinese and English tasks and even exceeds some larger models in benchmarks. These models feature the Transformer architecture with SwiGLU activation, attention QKV bias, group query attention, and a mix of sliding window and full attention mechanisms. They also include an advanced tokenizer for multiple languages and coding languages. Qwen's extensive vocabulary of over 150K tokens enhances compatibility with multiple languages, allowing for improved capabilities without expanding the vocabulary.

**Qwen-2-72B** Qwen2, is the newest series of large language models, developed by Alibaba, which surpasses its previous released Qwen1.5 series, yielding state-of-the-art performance across different benchmarks. Qwen2-72B-Instruct has an extended context length of up to 128K and is instruction-aligned with both supervised finetuning and direct preference optimization.

**Llama3-8B/70B** Meta's Llama 3 is the latest generation of llama family, release in April 18, 2024, featuring pretrained and instruction-fine-tuned versions with 8B and 70B parameters. Designed with a standard decoder-only transformer architecture, Llama 3 models demonstrate state-of-the-art performance across various industry benchmarks and show improved reasoning capabilities. Key enhancements include a tokenizer with a 128K token vocabulary for efficient language encoding and grouped query attention (GQA) for better inference efficiency.

Llama 3 models are pretrained on an extensive dataset of over 15T tokens from publicly available sources, including a significant increase in code and high-quality non-English data covering 30+ languages. This dataset is seven times larger than that used for Llama 2, ensuring superior model performance.

For instruction-tuning, Llama 3 employs a combination of supervised fine-tuning (SFT), rejection sampling, proximal policy optimization (PPO), and direct preference optimization (DPO). This approach, coupled with meticulously curated data and multiple quality assurance rounds, significantly enhances model alignment and response diversity.

**Llama3.1-8B/70B/405B** Introduced in July 2024, Llama 3.1 was pretrained on 15 trillion tokens of data from publicly available sources as well as over 25M synthetically generated examples. The intruction-tuned variants are post-trained using supervised fine-tuning (SFT) and reinforcement learning with human feedback (RLHF) to align with human preferences to guarantee safety and helpfulness. The Llama 3.1 405B demonstrates competitive performance across 150 benchmarks with leading foundation models, including GPT-4o and Claude 3.5 Sonnet.

### A.3.2 CODE-SPECIFIC LLMS

**CodeQwen1.5-7B** CodeQwen1.5-7B is based on Qwen1.5-7B. It is further trained on 3T tokens of code data, and it also includes group query attention (GQA) for efficient inference.

**DeepSeek-Coder-V2-16B-Instruct** DeepSeek-Coder-V2-16B-Instruct is an Mixture-of-Experts (MoE) code language model, that demonstrates comparable performance to GPT-4 Turbo in code-related tasks. Specifically, DeepSeek-Coder-V2 is continued to pre-train on intermediate checkpoint of DeepSeek-V2 with 6 trillion additional tokens, substantially enhance its reasoning capabilities in code and mathematical related tasks. DeepSeek-Coder-V2 supports 338 programming languages and has a context length of 128K.

**Codestral-22B-v0.1** Codestral-22B-v0.1 is the code-specific variant of Mistral-0.1-22B, it's trained on a diverse dataset of 80+ programming languages, including the most popular ones, such as Python, Java, C, C++, JavaScript, and Bash.

### A.3.3 GPT FAMILY

**GPT-3.5t** (GPT-3.5 Turbo) is a text only language model released by OpenAI on November 2022. The specific version of the model we are using is *gpt-3.5-turbo-0125*. It has a knowledge cutoff of September 2021 and a context window with length of 16K tokens.

**GPT-4t** (GPT-4 Turbo) is vision language model launched by OpenAI on March 2023. The specific version of the model we are using is *gpt-4-turbo-2024-04-09*. It has an updated knowledge cutoff of April 2023 and a context window with length of 128K tokens. It is more powerful than GPT-3.5.

**GPT-4o** (GPT-4 Omni) is a multimodal model released by OpenAI on May 2024, which support data types such as audio, vision, and text. The specific version of the model we are using is *gpt-4o-2024-05-13*. It has similar performance as GPT-4t on English text and code, but with significant improvement on non-English text, *i.e.*, over 50 languages. At the same time, it is able to reason with vision input. GPT-4o has knowledge up to October 2023 and supports context window with length of 128K tokens.

**GPT-4o mini** is a multimodal model released by OpenAI in July 2024, which is a more cost-efficient and smaller modal than GPT-4. It has a context window size of 128K tokens and the has knowledge up to October 2023.

### A.3.4 CLAUDE FAMILY

Claude is a multimodal, multilingual, proprietary model series developed by Anthropic. The Claude series includes different models: **Haiku**, the fastest and most lightweight model; **Sonnet**, the best balanced model between performance and speed; and **Opus**, the highest-performing model. We did not evaluate **Claude 3 Opus** because, in June 2024, Anthropic released **Claude 3.5 Sonnet**, the newest best-performing model.

Specifically, we are using *claude-3-5-sonnet-20240620* for **Claude 3.5 Sonnet**, *claude-3-sonnet-20240229* for **Claude 3 Sonnet**, and *claude-3-haiku-20240307* for **Claude 3 Haiku** for benchmark evaluation.

## B  HUMAN STUDY DETAILS

We ran a human study to verify the labels produced by GPT-4o for the benchmark over a subset of $500$ stimuli. We recruited $55$ participants from the crowdsourcing platform, Prolific [73]. Stimuli were batched into $10$ sets of $50$ stimuli each. Each participant was randomly assigned a batch of $50$ stimuli; stimuli were presented in a random shuffled. On each trial, participants saw the question, original image, and set of multiple choice options. Participants selected an option by clicking a button. We include an example screenshot of a trial in Figure 14. Participants were paid at a base rate of $12.50/hr. They were informed that they could receive a bonus up to $15/hr$ based on the amount of correct answers they achieved. All participants received the full bonus. Our study was approved by our institutional ethics review board, and all participants provided informed consent. We include the set of instructions and sample screenshots in Figures 15 and 16, respectively. We found high inter-annotator agreement (participants in the same batch had between $0.7 - 0.85$ Fleiss Kappa's alpha agreement, where higher implies higher agreement). We find that humans' mode response matched GPT-4o on 90% of the examples (450 of the 500 stimuli).

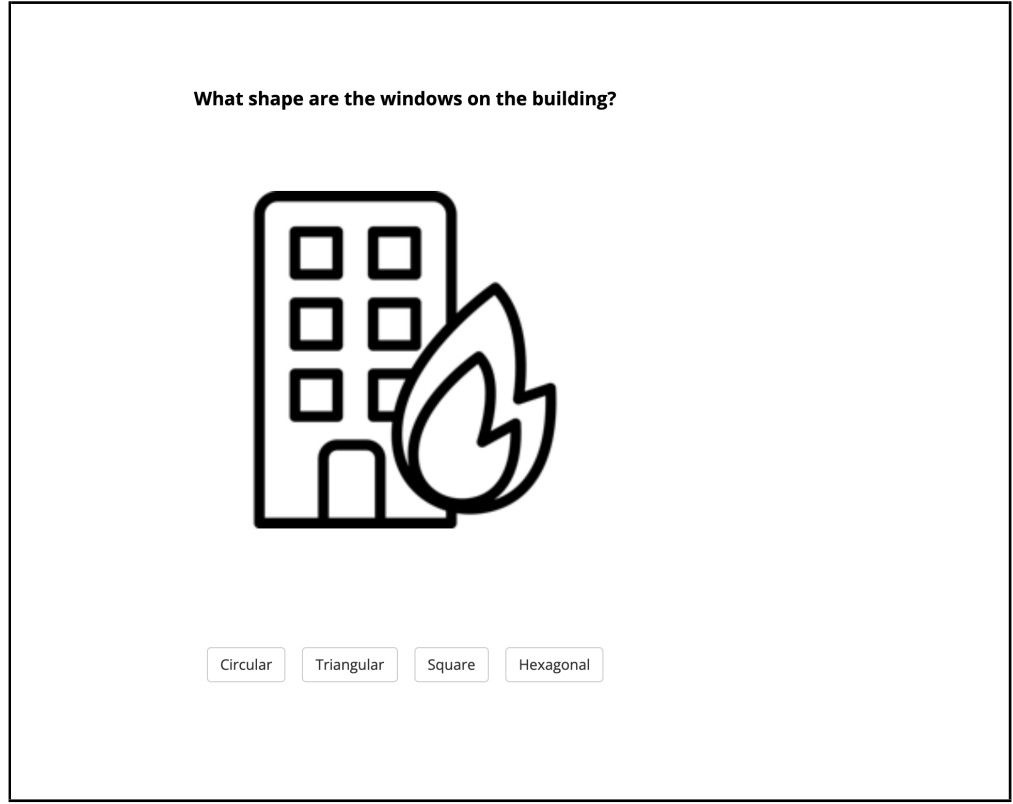

Figure 14: Example survey question.

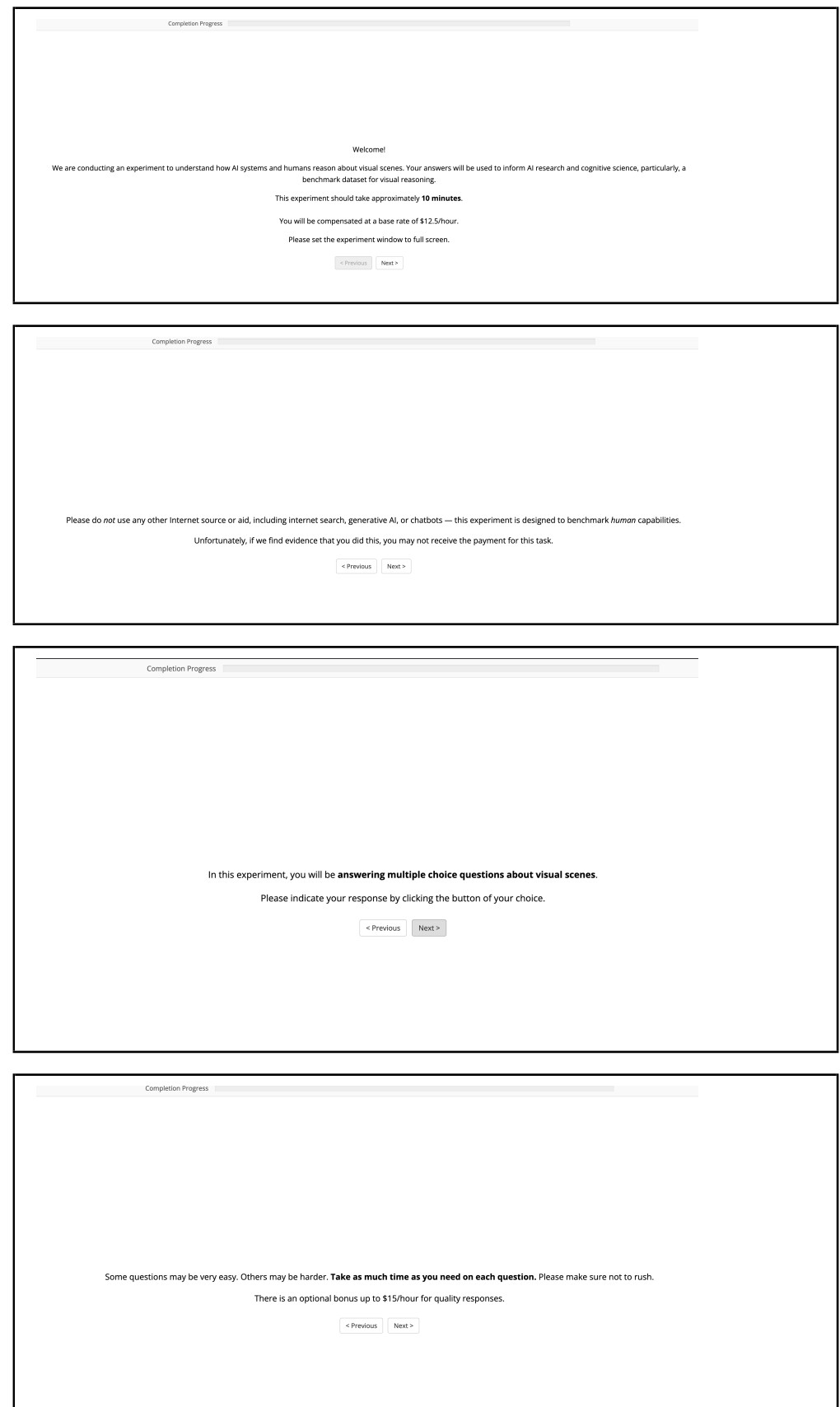

Figure 15: Experiment instructions.

Completion Progress

You will see a total of **50 questions**.

< Previous | Next >

Completion Progress

When you are ready, please click **"Next"** to complete a quick comprehension check, before moving on to the experiment.

Please make sure to window size is in full screen to properly view the questions.

< Previous | Next >

Completion Progress

Check your knowledge before you begin. If you don't know the answers, don't worry; we will show you the instructions again.

What will you be asked to do in this task?*

○ Answer multiple choice questions about visual reasoning tasks.
○ Draw sketches of visual scenes.
○ Provide captions for visual scenes.

How will you provide your final answer?*

○ Typing in text box.
○ Clicking a button indicating your choice.
○ Selecting from a dropdown menu.

Do you affirm that you will not use any other Internet source or aid, including internet search, or chatbots?*

○ I affirm that I will not use any other Internet source or aid, including internet search, or chatbots.
○ No, I do not affirm that I will not use any other Internet source or aid, including internet search, or chatbots. I might use them.

Continue

Completion Progress

Now you are ready to begin!

Please click **"Next"** to start the experiment.

Thank you for participating in our study!

< Previous | Next >

Figure 16: Experiment instructions (continued).

## C   SVG - Invariance Illustration

Our SVG - Invariance test is essential for testing whether a model has a fundamental understanding of the code, or it is able to pass the benchmark tests due to memorizing the SVG code samples, since we built our SVG-Bench using public available SVG datasets. In Figure 17 we see two SVG codes, illustrating two samples, that are semantically identical. The rotated sample is generated by ourself by applying a SE(2) transformation on the original sample (from *SVG Icons*). We can see that semantically these two samples are identical, the code changed drastically.

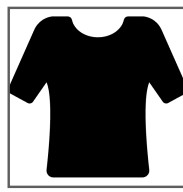

Original

```
<svg id="Capa_1" style="enable-background:new 0 0 298.667 298.667;" version="1.1" viewBox="0 0 298.667
298.667" x="0px" y="0px" xml:space="preserve">
 <g>
  <path
d="M0.604,134.717c-1.483,3.342-0.15,7.264,3.063,9.01l28.862,15.682c3.141,1.707,7.063,0.779,9.106-2.154l22.
406-32.165
c11.774,28.66,3.631,113.167-0.035,145.359c-0.367,3.219,0.658,6.442,2.817,8.858c2.159,2.416,5.246,3.792,8.48
6,3.792h148.047
c3.238,0,6.32-1.382,8.477-3.796s3.193-5.637,2.827-8.854c-3.666-32.192-11.809-116.698-0.035-145.359l22.406,
32.165   c2.043,2.933,5.965,3.861,9.106,2.154L295,143.727c3.213-1.746,4.546-5.667,3.063-9.01L255.015,37.71
c-4.193-9.437-12.08-16.759-21.829-20.249c-2.582-0.924-5.233-1.548-7.908-1.892l-25.618,0c-3.516,0-6.541,2.45
1-7.28,5.889
c-3.519,16.365-21.452,28.822-43.046,28.822c-21.594,0-39.527-12.456-43.046-28.821c-0.738-3.431-3.771-5.889
-7.28-5.889l-25.618,0
c-2.674,0.344-5.326,0.969-7.908,1.892c-9.749,3.49-17.636,10.813-21.829,20.249L0.604,134.717z" />
 </g>
</svg>
```

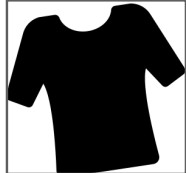

Rotated

```
<svg id="Capa_1" style="enable-background:new 0 0 298.667 298.667;" version="1.1" viewbox="0 0 298.667 298.667"
x="0px" xml:space="preserve" y="0px">
 <g>
  <path
d="M0.08497124091795172,157.01353292632658C-0.8841617683852974,160.53901646908758,1.017662534846778,164.
2189671915343,4.454721950394287,165.46736919970158L35.32910693826702,176.6795286536434 2C38.689164173223
62,177.9000799134501,42.42938657654092,176.39874571557027,44.013150638 8647,173.194369115529 07L61.38288566
434426,138.05310610258903C77.2909273289573,164.6417669638 8355,81.81484833500208.249.41956852025248,82.980
45135812532,281.7986634673397C83.09658635086157,285.03643471674286,84.58981594452071,288.0710054245401,8
7.08432092832022,290.13880180856813C89.57882591211973,292.2065981925961,92.8362 2538891045,293.1078715410
3343,96.04014623172823,292.62569809406875L242.43856259488945,270.5934967660 4694C245.64050570879192,270.1
116209570124,248.48251815751382,268.2863502260496,250.25624981090428,265.578 22891751 1S252.5748015178724 6,
259.52882101407823,251.7341265156018,256.40211179501784C243.31816928891686,225.11415750013606,222.689750
5443896,142.7610104740483,230.06734030978453,112.6669717578251L257.0105993609562,141.1393593061403C259.4
673354726304,143.73566161473153,263.4837658751966,144.0696598502683,266.3357551632589,141.91422785162442
L292.5435709937984,122.1114936365 5903C295.4609545831728,119.90678095857828,296.1955917430269,115.8310677
3959912,294.23160395194606,112.74599238290436L237.22648542993093,23.225576488 65834C231.67577218519597,1
4.517660782647283,222.78694574507878,8.450930401505076,212.6271279591 1045,6.450629535154889C209.93637120
566882,5.921169344197892,207.2220284941824,5.698636999131651,204.5256223459437,5.7565583573727 68L179.1928
9266961482,9.569002605626793C175.71604523648296,10.09225008696 2622,173.08948553555297,12.9661342421 5235,
172.8703543033994,16.475827462804972C171.82596264096188,33.18228825063885,155.94649417015597,48.16934330
1070896,134.59295507145015,51.38294003371189C113.2394159727443,54.5965367663529,93.65242062408873,44.9480
1158310621,87.73718423351656,29.288938671734655C86.49680439446237,26.005973113756937,83.13178147813346,2
4.026712198015574,79.6618560962052,24.54891794659561L54.329126419876 3,28.36136219484962C51.7360965840926
9,29.099473480995627,49.20663989557427,30.11218166251159,46.790751770695564,31.409154124852236C37.6696903
61213,36.31112717296749,30.960317665169526,44.72631677425315,28.21826474900425,54.68123971744809L0.084971
24091795172,157.01353292632658z"/>
 </g>
</svg>
```

Figure 17: Illustration of the SVG - Invariance test.

# D   MORE EXAMPLES IN SGP-BENCH

## D.1   SVG DATA

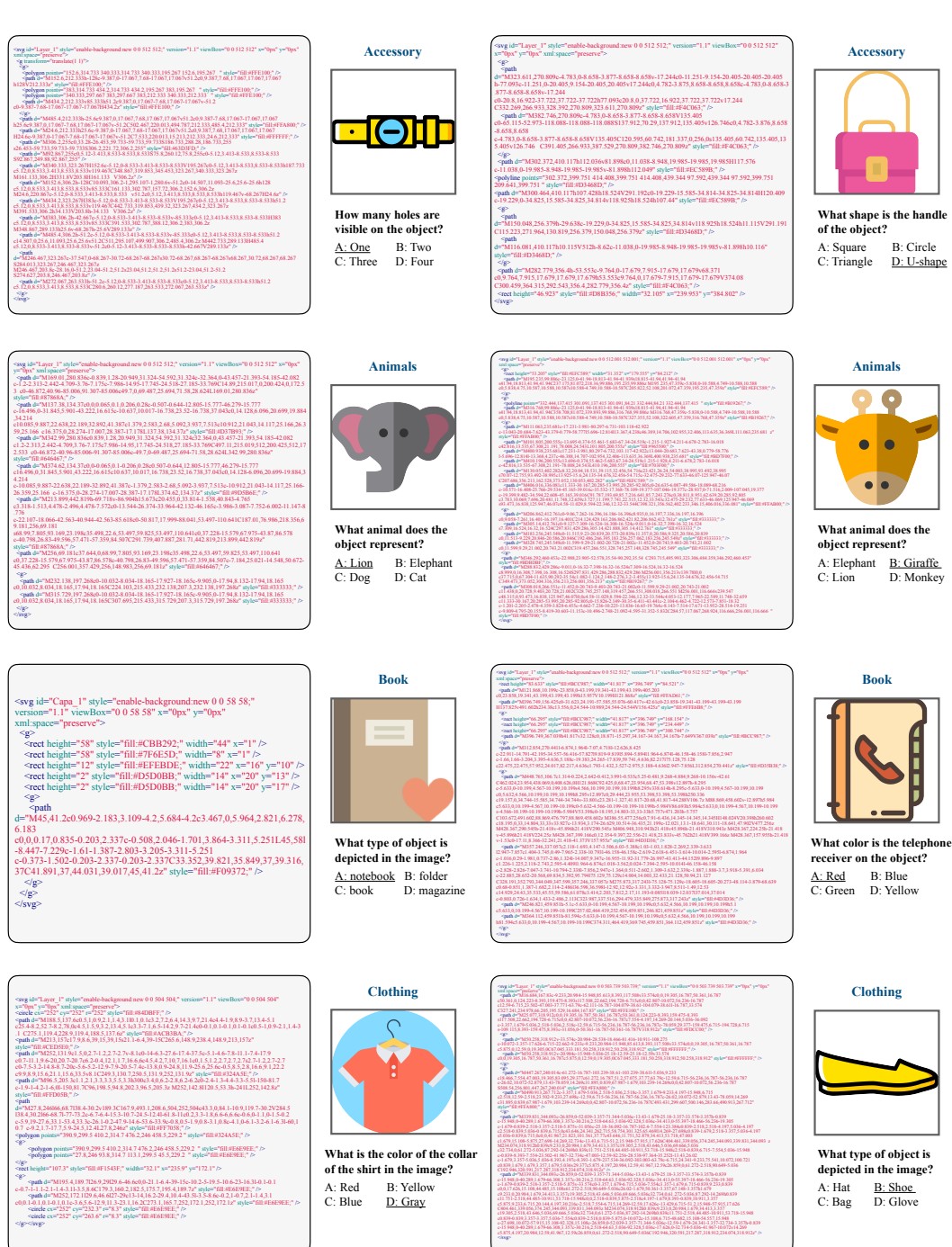

Figure 18: SVG examples in our SGP-Bench.

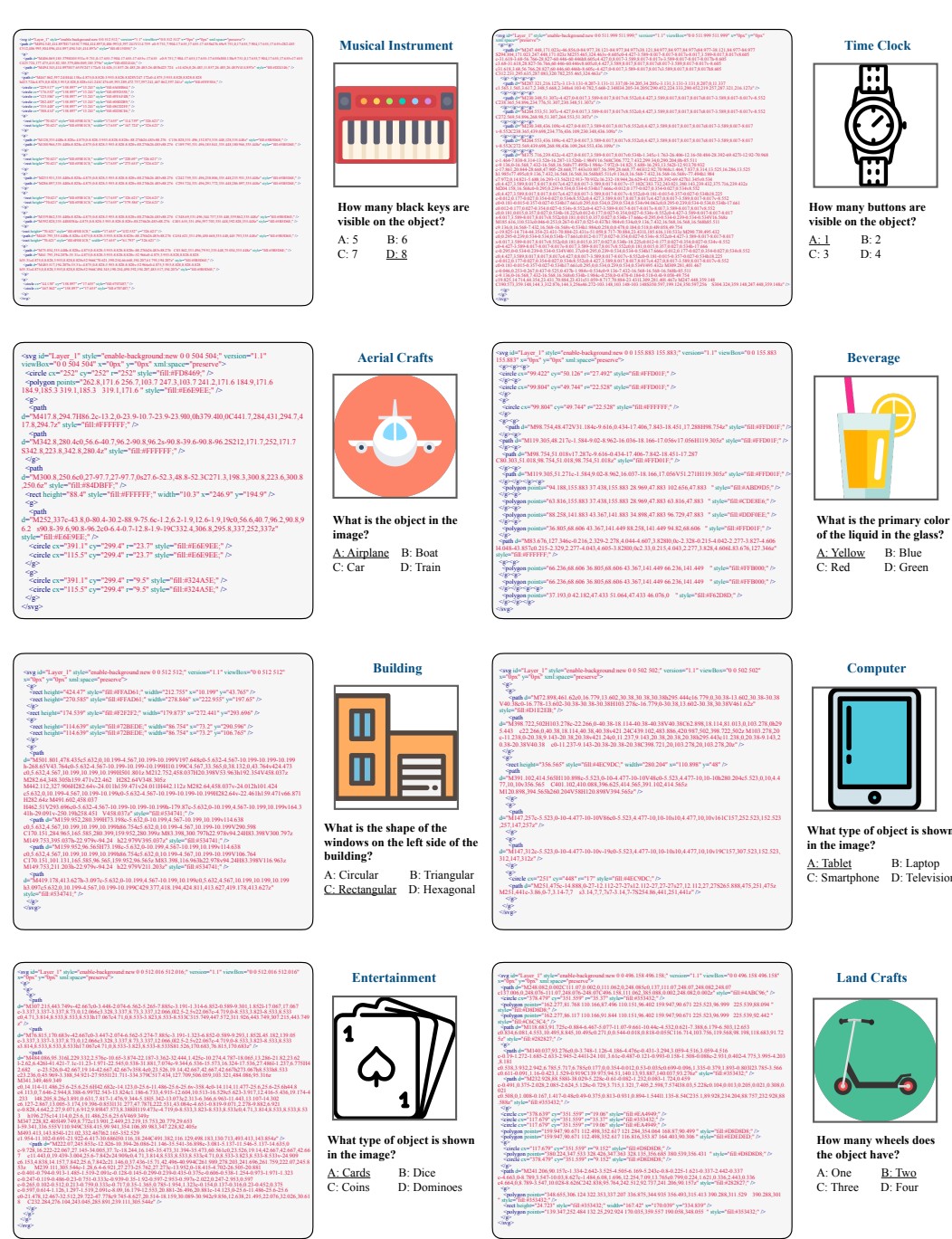

Figure 19: SVG examples in our SGP-Bench.

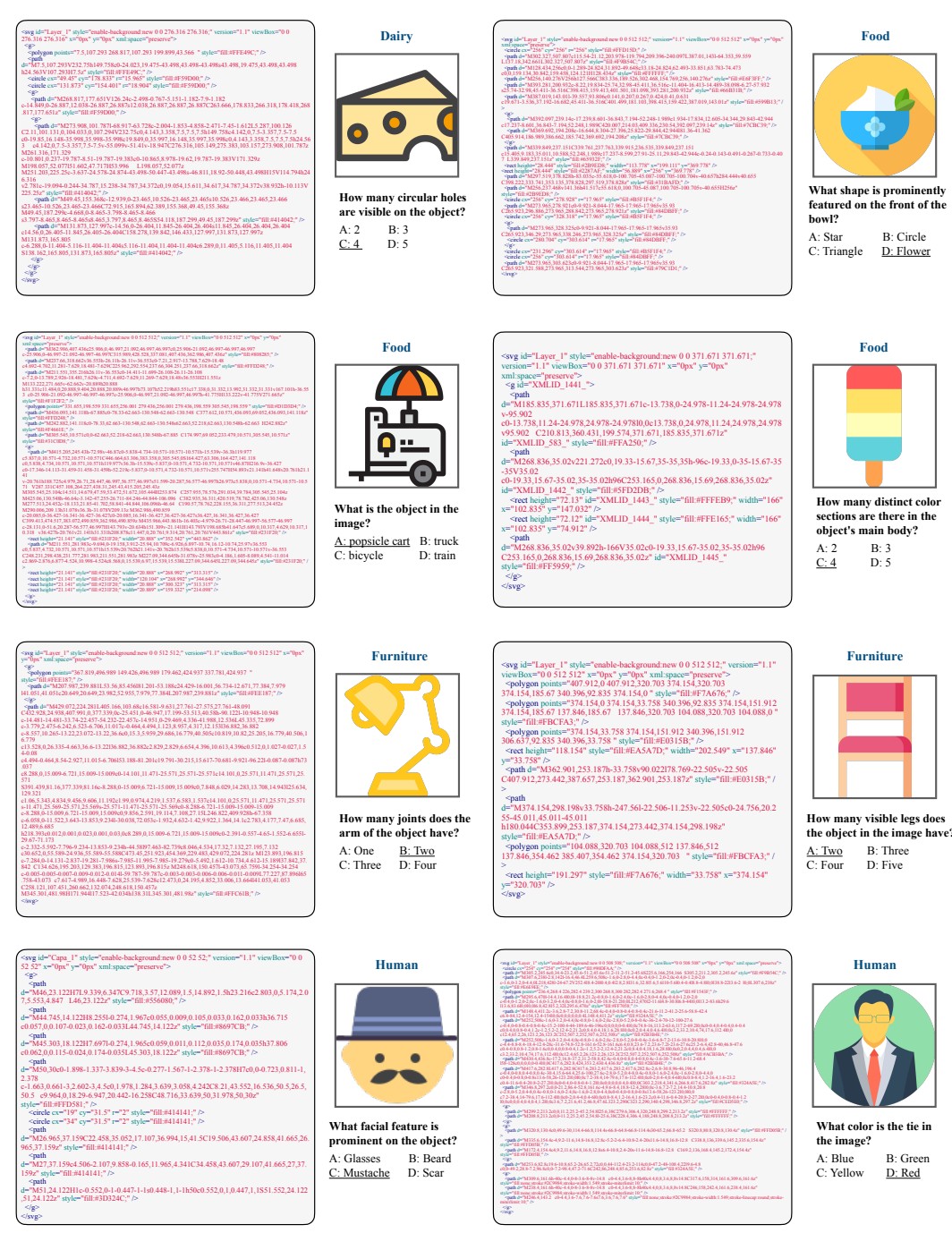

Figure 20: SVG examples in our SGP-Bench.

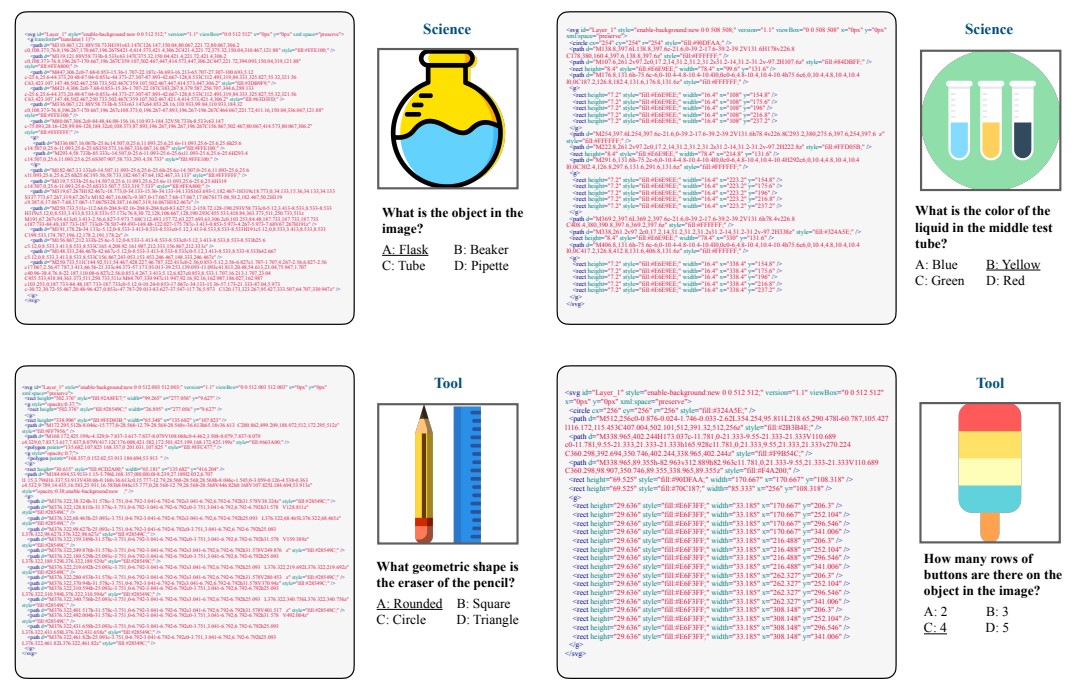

Figure 21: SVG examples in our SGP-Bench.

## D.2 CAD DATA

**SOL**;
**Arc**:(153,128,128,1);
**Line**:(153,169);
**Line**:(153,211);
**Arc**:(128,211,128,1);
**Line**:(128,169);
**Line**:(128,128);
**SOL**;
**Circle**:(140,128,6);
**SOL**;
**Circle**:(140,211,6);
**Ext**:(192,64,192,106,128,54,170,143,128,
**Newbody**,**One**-sided);
**EOS**

**1 Operation**

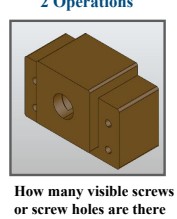

How many visible cylindrical sections does the CAD object have?

A: One   B: Two
C: Three   D: Four

---

**SOL**;
**Arc**:(130,126,64,1);**Line**:(221,126);**Arc**:(223,128,64,1);
**Line**:(223,176);**Arc**:(221,178,64,1);**Line**:(130,178);
**Arc**:(128,176,64,1);**Line**:(128,128);
**SOL**;
**Circle**:(139,138,3);
**SOL**;
**Circle**:(139,161,3);
**SOL**;
**Circle**:(176,161,10);
**SOL**;
**Circle**:(212,138,3);
**SOL**;
**Circle**:(212,161,3);
**Ext**:(192,64,192,34,128,62,189,207,128,**Newbody**,**One**-sided);
**SOL**;
**Arc**:(131,125,64,1);**Line**:(220,125);**Arc**:(223,128,64,1);
**Line**:(223,217);**Arc**:(220,220,64,1);**Line**:(131,220);
**Arc**:(128,217,64,1);**Line**:(128,128);
**SOL**;
**Circle**:(176,173,18);
**Ext**:(192,64,192,74,128,78,107,224,128,**Join**,**One**-sided);**EOS**

**2 Operations**

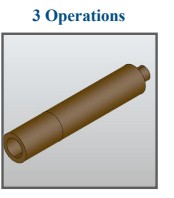

How many visible screws or screw holes are there on the CAD object?

A: Two   B: Three
C: Four   D: Five

---

**SOL**;
**Line**:(223,128);
**Line**:(223,153);
**Line**:(128,153);
**Line**:(128,128);
**Ext**:(192,64,192,32,128,83,174,157,128,
**Newbody**,**One**-sided);
**SOL**;
**Circle**:(176,128,48);
**Ext**:(192,64,192,39,99,106,21,12,128,
**Cut**,**One**-sided);
**EOS**

**2 Operations**

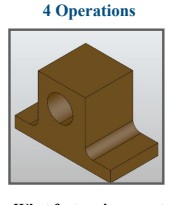

What type of feature is present on the side of the CAD object?

A: indentation   B: protrusion
C: hole   D: bump

---

**SOL**;
**Arc**:(131,125,64,1);**Line**:(211,125);**Arc**:(214,122,64,0);**Line**:(214,42);
**Arc**:(216,40,64,1);**Line**:(220,40);**Arc**:(223,42,64,1);**Line**:(223,132);
**Arc**:(220,135,64,1);**Line**:(131,135);**Arc**:(128,132,64,1);**Line**:(128,128);
**Ext**:(192,64,192,109,128,146,19,223,128,**Newbody**,**One**-sided);
**SOL**;
**Arc**:(131,125,64,1);**Line**:(211,125);**Arc**:(214,122,64,0);**Line**:(214,42);
**Arc**:(216,40,64,1);**Line**:(220,40);**Arc**:(223,42,64,1);**Line**:(223,132);
**Arc**:(220,135,64,1);**Line**:(131,135);**Arc**:(128,132,64,1);**Line**:(128,128);
**Ext**:(192,64,192,109,33,146,19,129,128,**Newbody**,**One**-sided);
**SOL**;
**Arc**:(129,127,64,1);**Line**:(222,127);**Arc**:(223,128,64,1);**Line**:(223,182);
**Arc**:(222,184,64,1);**Line**:(129,184);**Arc**:(128,182,64,1);**Line**:(128,128);
**SOL**;
**Circle**:(140,143,4);
**SOL**;
**Circle**:(140,167,4);
**SOL**;
**Arc**:(153,174,64,1);**Line**:(193,174);**Arc**:(194,173,64,0);**Line**:(194,133);
**Arc**:(196,131,64,1);**Line**:(198,131);**Arc**:(199,133,64,1);**Line**:(199,177);
**Arc**:(198,179,64,1);**Line**:(153,179);**Arc**:(152,177,64,1);**Line**:(152,176);
**SOL**;
**Circle**:(211,143,4);
**SOL**;
**Circle**:(211,167,4);
**Ext**:(192,64,192,99,33,127,38,129,128,**Join**,**One**-sided);
**EOS**

**3 Operations**

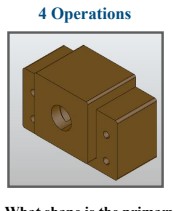

How many holes are visible on the flange of the CAD object?

A: Two   B: Three
C: Four   D: Five

---

**SOL**;
**Line**:(223,128);**Line**:(223,139);**Line**:(206,139);
**Arc**:(198,146,64,0);**Line**:(198,189);**Line**:(151,189);
**Line**:(151,146);**Arc**:(143,139,64,0);**Line**:(128,139);
**Line**:(128,128);
**Ext**:(192,64,192,69,128,148,202,128,
**Newbody**,**One**-sided);
**SOL**;
**Circle**:(176,128,47);
**Ext**:(192,64,192,119,54,189,40,4,128,
**Cut**,**One**-sided);
**SOL**;
**Circle**:(176,128,48);
**Ext**:(128,128,128,71,92,144,17,4,128,
**Cut**,**One**-sided);
**SOL**;
**Circle**:(176,128,48);
**Ext**:(128,128,128,198,90,144,16,4,128,
**Cut**,**One**-sided);
**EOS**

**4 Operations**

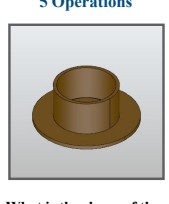

What feature is present at the base of the CAD object?

A: gears   B: surface
C: fins   D: recess

---

**SOL**;
**Circle**:(176,128,48);
**SOL**;
**Circle**:(176,128,24);
**Ext**:(192,64,192,116,128,128,24,43,128,
**Newbody**,**One**-sided);
**SOL**;
**Circle**:(176,128,48);
**Ext**:(192,64,192,116,128,128,24,166,128,
**Newbody**,**One**-sided);
**SOL**;
**Circle**:(176,128,48);
**Ext**:(192,64,192,120,90,128,16,99,128,
**Cut**,**One**-sided);
**SOL**;
**Circle**:(176,128,48);
**Ext**:(192,192,192,134,128,128,12,224,128,
**Join**,**One**-sided);
**EOS**

**4 Operations**

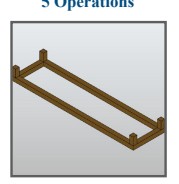

What shape is the primary body of the CAD object?

A: Square   B: Triangle
C: Cylinder   D: Sphere

---

**SOL**;
**Circle**:(176,128,48);
**SOL**;
**Circle**:(176,128,29);
**Ext**:(128,128,128,32,128,128,192,135,128,**Newbody**,**One**-sided);
**SOL**;
**Circle**:(176,128,47);
**SOL**;
**Circle**:(176,128,5);
**Ext**:(128,128,128,75,128,128,106,135,128,**Join**,**One**-sided);
**SOL**;
**Circle**:(176,128,48);
**SOL**;
**Circle**:(176,128,43);
**Ext**:(128,128,128,69,128,128,118,135,128,**Join**,**One**-sided);
**SOL**;
**Circle**:(176,128,48);
**SOL**;
**Circle**:(176,128,43);
**Ext**:(128,128,128,69,128,128,118,200,128,**Join**,**One**-sided);
**SOL**;
**Circle**:(176,128,48);
**SOL**;
**Circle**:(176,128,29);
**Ext**:(128,128,128,32,128,128,192,135,128,**Join**,**One**-sided);
**EOS**

**5 Operations**

What is the shape of the base of the CAD object?

A: Triangle   B: Square
C: Rectangle   D: Circle

---

**SOL**;
**Line**:(223,128);**Line**:(223,153);**Line**:(128,153);
**Line**:(128,128);**SOL**;**Line**:(220,131);**Line**:(220,150);
**Line**:(131,150);**Line**:(131,131);
**Ext**:(128,128,128,128,128,128,96,131,128,**Newbody**,**One**-sided);
**SOL**;
**Line**:(223,128);**Line**:(223,223);**Line**:(128,223);
**Line**:(128,128);
**Ext**:(128,128,128,128,150,131,3,134,128,**Join**,**One**-sided);
**SOL**;
**Line**:(223,128);**Line**:(223,223);**Line**:(128,223);
**Line**:(128,128);
**Ext**:(128,128,128,128,128,131,3,134,128,**Join**,**One**-sided);
**SOL**;
**Line**:(223,128);**Line**:(223,223);**Line**:(128,223);
**Line**:(128,128);
**Ext**:(128,128,128,221,150,131,3,134,128,**Join**,**One**-sided);
**SOL**;
**Line**:(223,128);**Line**:(223,223);**Line**:(128,223);
**Line**:(128,128);
**Ext**:(128,128,128,221,128,131,3,134,128,**Join**,**One**-sided);
**EOS**

**5 Operations**

How many vertical supports does the CAD object have?

A: Two   B: Four
C: Six   D: Eight

Figure 22: CAD examples (3D) in our SGP-Bench.

**6 Operations**

```
SOL;
Circle:(176,128,48);
SOL;
Circle:(176,128,32);
SOL;
Ext:(192,64,192,101,128,128,54,149,128,Newbody,Symmetric);
SOL;
Arc:(181,128,128,1);Line:(181,197),Arc:(128,197,128,1);
Line:(128,128);
SOL;
Arc:(166,128,128,1);Line:(166,197);Arc:(143,197,128,1);
Line:(143,128);
Ext:(192,64,192,113,128,170,54,137,128,Join,Symmetric);
SOL;
Arc:(223,146,15,0);Arc:(135,177,27,0);Arc:(128,128,85,0);
Ext:(192,64,192,118,128,153,10,137,128,Join,Symmetric);
SOL;
Arc:(223,110,15,0);Arc:(216,158,85,0);Arc:(128,128,27,0);
Ext:(192,64,192,128,128,155,10,137,128,Join,Symmetric);
SOL;
Arc:(174,165,24,0);Arc:(177,184,85,1);
Arc:(156,223,33,0);Arc:(128,128,26,0);
Ext:(192,64,192,106,128,143,25,135,128,Join,Symmetric);
SOL;
Arc:(133,95,85,1);Arc:(211,33,24,0);Arc:(164,193,26,0);
Arc:(128,128,33,0);
Ext:(192,64,192,137,128,158,15,135,128,Join,Symmetric);EOS
```

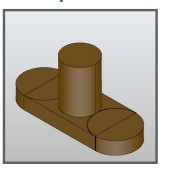

**What type of geometric feature is the central hole of the CAD object?**

A: Square B: Hexagonal
C: Circular D: Triangular

**6 Operations**

```
SOL;
Arc:(130,126,64,1);Line:(221,126);Arc:(223,128,64,1);
Line:(223,176);Arc:(221,178,64,1);Line:(130,178);
Arc:(128,176,64,1);Line:(128,128);
SOL;
Circle:(139,138,3);
SOL;
Circle:(139,161,3);
SOL;
Circle:(176,161,10);
SOL;
Circle:(212,138,3);
SOL;
Circle:(212,161,3);
Ext:(192,64,192,34,128,62,189,207,128,Newbody,One-sided);
SOL;
Arc:(131,125,64,1);Line:(220,125);Arc:(223,128,64,1);
Line:(223,217);Arc:(220,220,64,1);Line:(131,220);
Arc:(128,217,64,1);Line:(128,128);
SOL;
Circle:(176,173,18);
Ext:(192,64,192,74,128,78,107,224,128,Join,One-sided);EOS
```

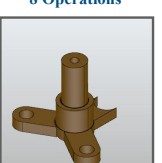

**What is the primary shape of the CAD object?**

A: Circular B: Triangular
C: Rectangular D: L-shaped

**7 Operations**

```
SOL;
Arc:(134,122,64,1);Line:(217,122);Arc:(223,128,64,1);
Line:(223,198);Arc:(217,204,64,1);Line:(134,204);
Arc:(128,198,64,1);Line:(128,128);
Ext:(192,64,192,32,128,57,192,82,128,Newbody,One-sided);
SOL;
Line:(223,128);Line:(223,220);Line:(128,220);
Line:(128,128);
Ext:(192,64,192,43,128,46,169,174,128,Join,One-sided);
SOL;
Circle:(176,128,47);
Ext:(192,192,214,174,194,11,14,128,Cut,One-sided);
SOL;
Circle:(176,128,47);
Ext:(192,192,54,174,194,11,14,128,Cut,One-sided);
SOL;
Circle:(176,128,47);
Ext:(192,192,214,174,62,11,14,128,Cut,One-sided);
SOL;
Circle:(176,128,47);
Ext:(192,192,54,174,62,11,14,128,Cut,One-sided);
SOL;
Line:(223,128);Line:(223,216);Line:(128,216);Line:(128,128);
Ext:(192,192,190,174,71,123,91,128,Cut,One-sided);
EOS
```

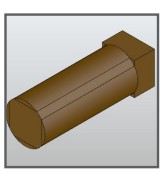

**What shape primarily makes up the body of the CAD object?**

A: Cylinder B: Sphere
C: Cube D: Cone

**7 Operations**

```
SOL;
Line:(128,223);Arc:(128,128,128,1);
Ext:(128,128,128,122,83,128,45,142,128,Newbody,One-sided);
SOL;
Arc:(128,223,128,1);Line:(128,128);
Ext:(128,128,128,122,83,128,45,142,128,Join,One-sided);
SOL;
Line:(223,128);Arc:(223,182,128,0);
Line:(128,182);Arc:(128,128,128,0);
Ext:(128,128,128,122,83,128,80,142,128,Join,One-sided);
SOL;
Arc:(128,223,128,1);Line:(128,128);
Ext:(128,128,128,201,83,128,45,142,128,Join,One-sided);
SOL;
Line:(128,223);Arc:(128,128,128,1);
Ext:(128,128,128,201,83,128,45,142,128,Join,One-sided);
SOL;
Circle:(176,128,48);
Ext:(128,128,128,145,105,142,34,176,128,Join,One-sided);
SOL;
Line:(223,128);Line:(223,223);Line:(128,223);Line:(128,128);
Ext:(255,128,255,149,118,128,26,118,128,Cut,One-sided);
EOS
```

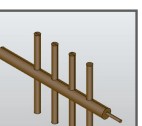

**What shape is the primary feature of the CAD object?**

A: Square B: Triangle
C: Cylinder D: Cone

**8 Operations**

```
SOL;
Line:(223,128);Line:(223,207);
Line:(128,207);Arc:(128,128,128,0);
Ext:(128,128,128,76,84,128,105,145,128,Newbody,One-sided);
SOL;Line:(128,156);Arc:(128,194,128,0);
Line:(128,223);Arc:(128,128,128,1);
Ext:(128,128,128,76,84,128,87,145,128,Join,One-sided);
SOL;Arc:(128,223,128,1);Line:(128,194);
Arc:(128,156,128,0);Line:(128,128);
Ext:(128,128,128,76,84,128,87,145,128,Join,One-sided);
SOL;Line:(128,156);Arc:(128,194,128,0);
Line:(128,223);Arc:(128,128,128,1);
Ext:(128,128,128,76,84,128,87,198,128,Join,One-sided);
SOL;Arc:(128,223,128,1);Line:(128,194);
Arc:(128,156,128,0);Line:(128,128);
Ext:(128,128,128,76,84,128,87,198,128,Join,One-sided);
SOL;Line:(176,128);Line:(223,128);Line:(223,147);
Line:(128,147);Line:(128,128);
Ext:(192,128,192,180,84,128,87,111,128,Join,One-sided);
SOL;Line:(223,128);Line:(223,204);
Line:(128,204);Line:(128,128);
SOL;Circle:(185,166,19);
Ext:(192,128,192,180,84,93,44,111,128,Join,One-sided);
SOL;Line:(223,128);Line:(223,204);
Line:(128,204);Line:(128,128);
SOL;Circle:(166,166,19);
Ext:(192,128,192,180,84,93,44,111,128,Join,One-sided);
EOS
```

**What is the shape of the main vertical feature in the CAD object?**

A: Cube B: Sphere
C: Cylinder D: Cone

**8 Operations**

```
SOL;
Circle:(176,128,48);
SOL;
Circle:(176,128,24);
Ext:(192,64,192,116,128,128,24,43,128,
Newbody,One-sided);
SOL;
Circle:(176,128,48);
Ext:(192,64,192,116,128,128,24,166,128,
Newbody,One-sided);
SOL;
Circle:(176,128,48);
Ext:(192,64,192,120,90,128,16,99,128,
Cut,One-sided);
SOL;
Circle:(176,128,48);
Ext:(192,192,192,134,128,128,12,224,128,
Join,One-sided);
EOS
```

**How many arms does the CAD object have extending from its central body?**

A: One B: Two
C: Three D: Four

**9 Operations**

```
SOL;
Line:(223,128);Line:(223,223);
Line:(128,223);Line:(128,128);
Ext:(192,64,192,46,128,194,27,145,128,Newbody,One-sided);
SOL;Line:(223,128);Arc:(128,128,50,1);
Ext:(192,64,192,50,111,221,19,206,128,Join,One-sided);
SOL;Line:(223,128);Arc:(128,128,50,1);
Ext:(192,64,192,46,111,198,19,206,128,Join,One-sided);
SOL;Arc:(223,128,50,1);Line:(128,128);
Ext:(192,64,192,50,111,194,19,206,128,Join,One-sided);
SOL;Arc:(128,223,50,1);Line:(128,128);
Ext:(192,64,192,73,111,198,19,206,128,Join,One-sided);
SOL;Arc:(142,114,14,1);Line:(209,114);Arc:(223,128,14,1);
Line:(223,195);Arc:(209,209,14,1);Line:(142,209);
Arc:(128,195,14,1);Line:(128,128);
Ext:(192,64,192,46,111,198,27,206,128,Join,One-sided);
SOL;Circle:(176,128,48);
Ext:(128,128,128,54,89,221,11,111,183,Cut,Two-sided);
SOL;Circle:(176,128,48);
Ext:(128,128,128,54,67,221,11,111,183,Cut,Two-sided);
SOL;Circle:(176,128,48);
Ext:(128,128,128,54,45,221,11,111,183,Cut,Two-sided);EOS
```

**What shape is the main body of the CAD object?**

A: Square B: Cylinder
C: Sphere D: Cone

**10 Operations**

```
SOL;
Line:(223,128);Line:(223,153);Line:(128,153);
Line:(128,128);SOL;Line:(220,131);Line:(220,150);
Line:(131,150);Line:(131,131);
Ext:(128,128,128,128,128,96,131,128,Newbody,One-sided);
SOL;
Line:(223,128);Line:(223,223);Line:(128,223);
Line:(128,128);
Ext:(128,128,128,150,131,3,134,128,Join,One-sided);
SOL;
Line:(223,128);Line:(223,223);Line:(128,223);
Line:(128,128);
Ext:(128,128,128,128,131,3,134,128,Join,One-sided);
SOL;
Line:(223,128);Line:(223,223);Line:(223,223);
Line:(128,128);
Ext:(128,128,128,221,150,131,3,134,128,Join,One-sided);
SOL;
Line:(223,128);Line:(223,223);Line:(128,223);
Line:(128,128);
Ext:(128,128,128,221,128,131,3,134,128,Join,One-sided);
EOS
```

**How many cylindrical rods are perpendicular to the main horizontal cylinder in the CAD image?**

A: 3 B: 4
C: 5 D: 6

Figure 23: CAD examples (3D) in our SGP-Bench.

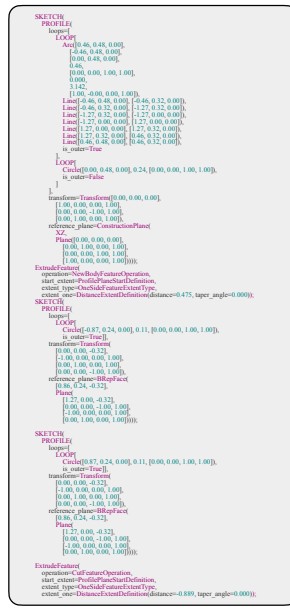

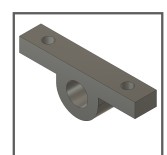

**How many holes are visible on the CAD object?**

A: One    B: Two

C: Three    D: Four

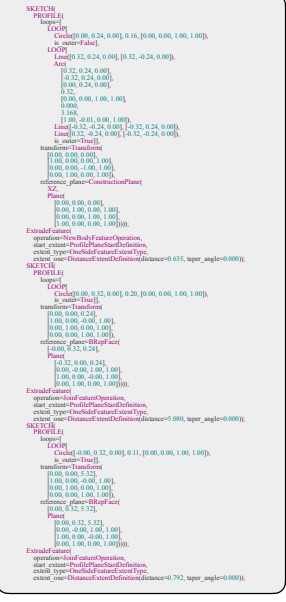

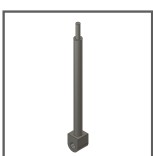

**How many holes are visible on the CAD object?**

A: One    B: Two

C: Three    D: Four

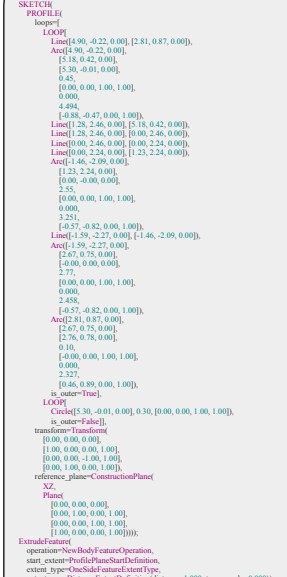

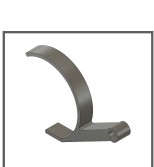

**What is the orientation of the curved beam in relation to the base?**

A: Perpendicular    B: Parallel

C: Diagonal    D: Inclined

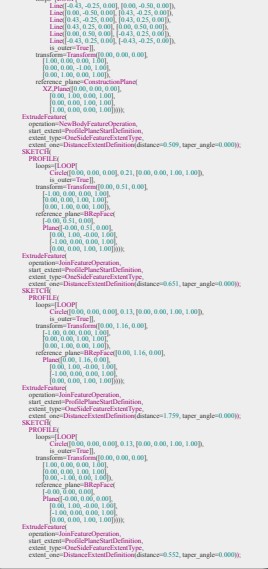

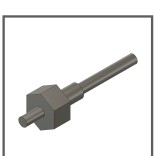

**What is the orientation of the smaller cylinder relative to the larger cylinder in the CAD object?**

A: Perpendicular    B: Parallel

C: Acute angle    D: Obtuse angle

Figure 24: CAD examples (3D$_{complex}$) in our SGP-Bench.

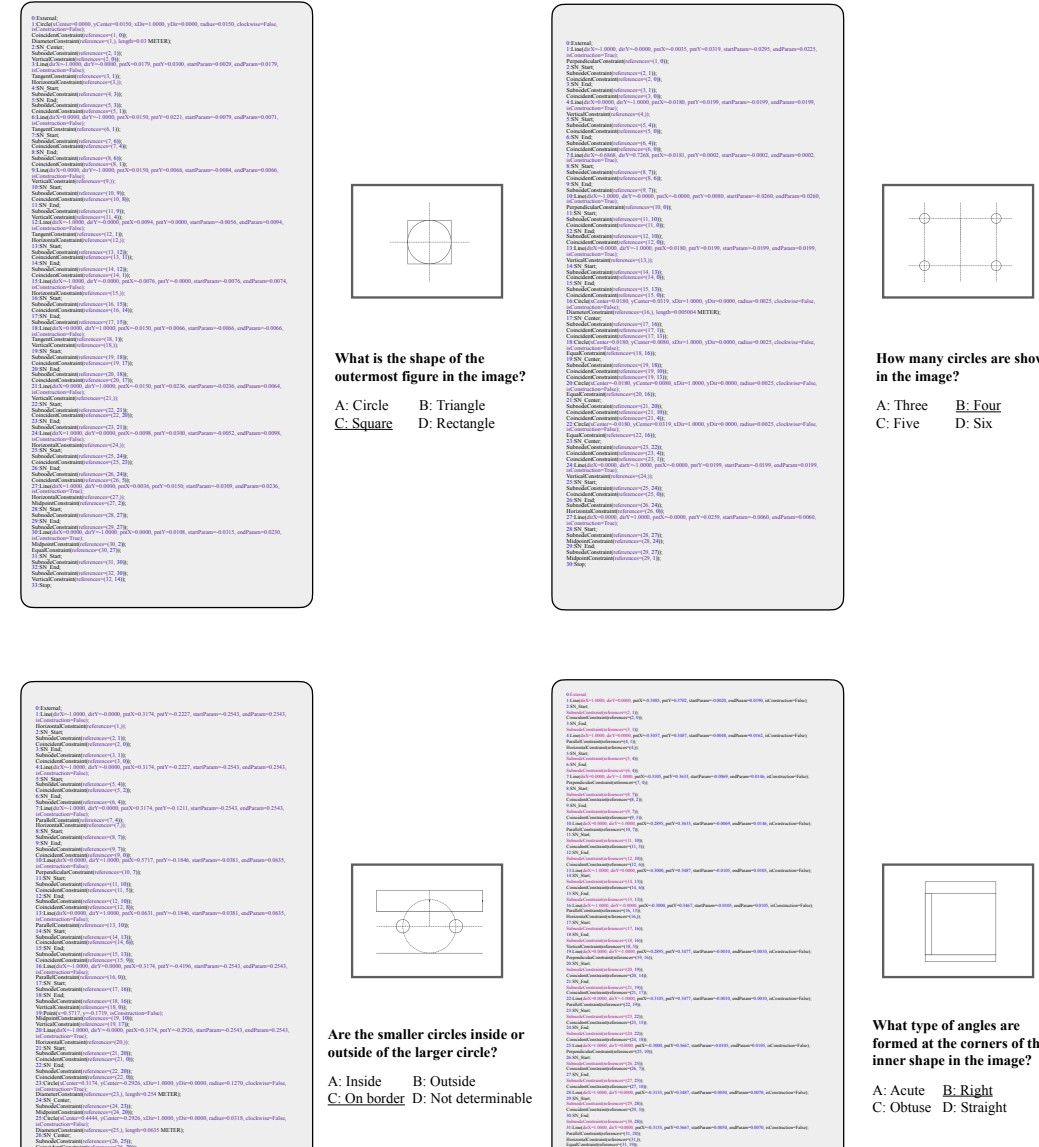

Figure 25: CAD examples (2D) in our SGP-Bench.

## D.3  Symbolic Instruction-following Data (SVG)

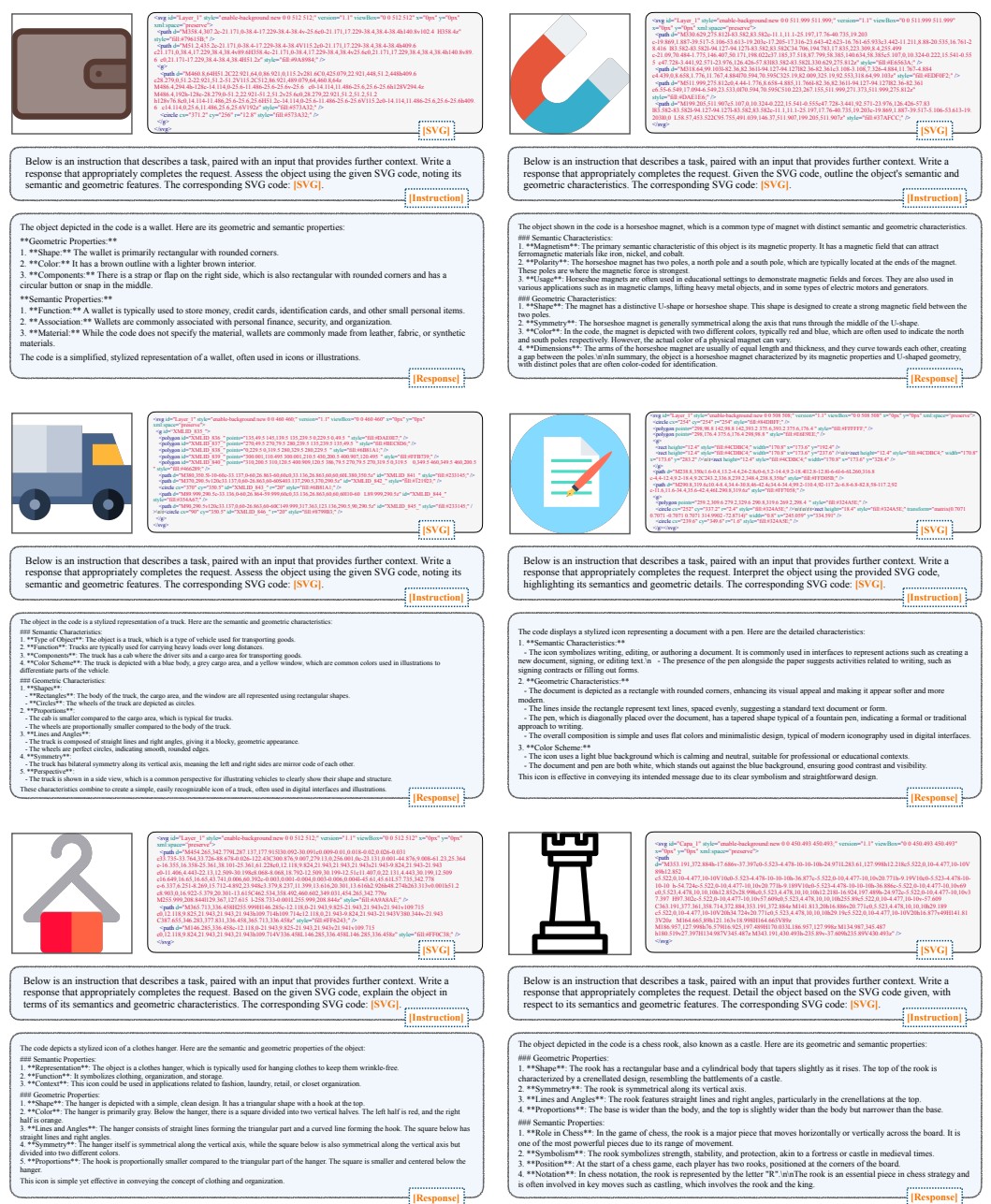

Figure 26: SVG examples of our symbolic instruction-following data.

# E  DETAILS AND MORE RESULTS OF SYMBOLIC INSTRUCTION TUNING

## E.1  IMPLEMENTATION DETAILS

We use the *unsloth*[9] framwork to finetune the base models Llama3-8b-instruct and Gemma-1.1-7b-it. For both models, we use the exact same training setting: we finetune the base models with LoRA [36] on 1 NVIDIA H100 80GB gpu with learning rate 2e-4, batch size of 2 and for 1 epoch.

We use the *PEFT*[10] framework to test different fine-tuning methods when performing SIT. We choose two common fine-tuning methods LoRA [36] and orthogonal finetuning [75, 60] to fine-tune the base model Llama3.1-8b-Instruct. For both fine-tuning methods, we train on 8 NVIDIA H100 80GB gpus with learning rate 1e-4, per device batch size of 1 and for 1 epoch.

As introduced in Section 6, we also use *PEFT* to test if SIT can improve generic instruction tuning, by mixing our curated SIT data into the publicly available instruction tuning dataset *open-instruct*[11]. We use LoRA to fine-tune the base model Llama3.1-8b on 8 NVIDIA H100 80GB gpus with learning rate 1e-4, per device batch size of 1 and for 1 epoch. We test mixing with different SIT data splits, including 10K, 25K, 40K, 55K, and 72K. For example, Open-Instruct-SIT-10K, Open-Instruct-rev-SIT-10K and Open-Instruct-mixed-SIT-10K are constructed by mixing Open-Instruct with SIT-10K, rev-SIT-10K and mixed-SIT-10K. More specifically, rev-SIT-10K is constructed from SIT-10K according to Figure 12, while mixed-SIT-10K uniformly samples exactly 5K of the SIT instruction-following pairs and convert them to rev-SIT, while the rest 5K is kept unchanged. The best result is reported in the Table 4. We employ the widely-used *lm-evaluation-harness*[12] to obtain the results on a variety of LLM benchmarks.

---

[9]https://github.com/unslothai/unsloth
[10]https://github.com/huggingface/peft
[11]https://huggingface.co/datasets/VMware/open-instruct
[12]https://github.com/EleutherAI/lm-evaluation-harness

### E.2 MORE EXPERIMENTS IN SYMBOLIC INSTRUCTION TUNING

We additionally provide an ablation study of using different-size SIT data to finetune the base LLMs and measure their performance after SIT on the SGP-Bench. We uniformly sample 72k SVG programs from the *SVG Icons* dataset to build an instruction-following dataset using the text prompt examples in F.1 to query GPT-4v. The SIT-25k dataset is built by choosing the samples with the shortest code length out of the original 72k instruction following pairs. The SIT-10k dataset is a subset of SIT-25k, by uniformly sampling from the SIT-25k dataset. For SIT-40k and SIT-55k, we additionally sample more data with short code length from the *SVG Icons* dataset and mix it with SIT-25k. In this way, we can ensure that the smaller SIT dataset is always a subset of the bigger one.

We use the LLM-based evaluation because we noticed that after SIT, the generic instruction-following ability of the finetuned model degerates compared to the original model. We want to eliminate the cases where the finetuned model answers the questions correctly but do not follow the answer template, so that a matching-based evaluation will fail to extract meaningful answers. The results are shown in the Table 6. We notice that for Llama3-8B-instruct, the SIT will improve the generic semantic understanding up-to some SIT data size, afterwards the performance degenerates, while for Gemma-1.1-7b-it, the overall semantic understanding improves significantly without noticeable degeneration.

| Dataset Size | Llama3-8B | Gemma-7B |
|---|---|---|
| Original | 43.20 | 39.33 |
| SIT-10k | **48.16** (+4.96) | **45.60** (+6.27) |
| SIT-25k | **51.43** (+8.23) | **46.87** (+7.54) |
| SIT-40k | **45.62** (+2.42) | **45.21** (+5.88) |
| SIT-55k | 40.99 (-2.21) | **47.28** (+7.95) |

Table 6: Ablation study of studying the effect of different-sized SIT data on the model's performance on the SPG-Bench (SVG-Understanding) using LLM-based answer extraction.

We also conducted an ablation study to determine whether the finetuning method affects the SIT results. Our findings indicate that the enhancement in the model's ability to understand symbolic programs is independent of the finetuning approach. Both OFT and LoRA significantly improve the model's understanding of symbolic programs, as shown in Table 7.

| Dataset Size | LoRA | OFT |
|---|---|---|
| Llama 3.1-8B* | 46.7 | 46.7 |
| SIT-10k | **47.9** (+1.2) | **48.0** (+1.3) |
| SIT-25k | **49.8** (+3.1) | **50.3** (+3.6) |
| SIT-40k | **51.0** (+4.3) | **51.2** (+4.5) |
| SIT-55k | **51.3** (+4.6) | **51.4** (+4.7) |

Table 7: Ablation study of studying the effect of different-sized SIT data on the model's performance on the SPG-Bench (SVG-Understanding) using LLM-based answer extraction and different fine-tuning methods. * The value differs from the value in the main table, because we use LLM-based evaluation to guarantee consistency.

# F  TEXT PROMPT TEMPLATE

## F.1  TEMPLATE FOR BENCHMARK CONSTRUCTION

We randomly sample from the following 20 prompts to generate the Symbolic Instruction Tuning (SIT) data:

```
"Describe in detail the semantic or geometric characteristics of
the object shown in the image."
"Offer a detailed description of the geometric or semantic
attributes of the object in this image."
"Can you provide a detailed account of the geometric or semantic
features of the object in the image?"
"Give a comprehensive description of the semantic or geometric
properties of the object depicted in the image."
"Elaborate on the geometric or semantic features of the object
in the image."
"Provide an in-depth description of the semantic or geometric
aspects of the object shown in the image."
"Detail the semantic or geometric features of the object in
the image."
"Explain in detail the semantic or geometric characteristics
of the object displayed in the image."
"Could you detail the geometric or semantic features of the object
in the image?"
"I need a detailed description of the geometric or semantic
attributes of the object in the image."
"Please describe the semantic or geometric features of the object
in the image comprehensively."
"Provide a thorough description of the geometric or semantic
properties of the object in this image."
"Can you elaborate on the semantic or geometric features of the
object in the image?"
"Describe precisely the semantic or geometric characteristics
of the object shown in the image."
"Give a detailed explanation of the geometric or semantic
features of the object in the image."
"Offer a complete description of the semantic or geometric
aspects of the object in the image."
"Detail the geometric or semantic properties of the object
depicted in the image."
"Explain the semantic or geometric features of the object in the
image in detail."
"Provide a detailed analysis of the geometric or semantic
features of the object in this image."
"Elaborate on the semantic and geometric characteristics of the
object shown in the image."
```

We use the following prompt to query GPT to construct the SGP-Bench (SVG) question-answer pairs.

```
Given the image, contruct in total 4 multiple-choice question-
answer pairs, with answer choices A, B, C, D, that concentrate
on the semantics or geometric features of the object in the img.
The first three questions are random.
The forth question should ask about the semantic of the
whole object.
Note: the format of the question-answer pairs should be
as follows:
===
Question: What is the capital of Germany?
Options: A) Rome; B) Beijing; C) Paris; D) Berlin
Answer: D
===
Question: What is the color of the sky on a clear day?
Options: A) Gray; B) Blue; C) Orange; D) Green
Answer: B
===
```

We use the following prompt to query GPT to construct the SGP-Bench (CAD) question-answer pairs:

```
Construct five multiple-choice question-answer pairs, with
answer choices A, B, C, D, that concentrate on the geometry
of the CAD object in the image.
Note: the format of the question-answer pairs should be
as follows:
===
Question: How did Spider-Man get his powers?
Options: A) Bitten by a radioactive spider; B) Born with them;
C) Military experiment gone awry; D) Woke up with them after
a strange dream
Answer: D
===
Question: What is the color of the sky on a clear day?
Options: A) Gray; B) Blue; C) Orange; D) Green
Answer: B
===
```

We randomly sample from the following 20 prompts to construct the questions for the SGP-MNIST benchmark. We do not need to query GPT because we have the ground truth label for every SGP-MNIST code.

```
"What number between 0 and 9 is shown in this picture?"
"Identify the digit from 0 to 9 depicted in this image."
"Which number from 0 through 9 is illustrated in this image?"
"Can you tell which digit (0-9) this image represents?"
"What is the digit, from 0 to 9, that appears in this image?"
"Determine the digit between 0 and 9 displayed in this image."
"Spot the digit (0-9) that this image portrays."
"Which of the digits 0-9 does this image illustrate?"
"Recognize the digit from 0-9 shown in this picture."
"From 0 to 9, what digit is shown here in this image?"
"What single digit from 0-9 is presented in this image?"
"Specify the digit (0-9) that is represented by this image."
"What digit, ranging from 0 to 9, does this image show?"
"Identify which one of the digits 0-9 is depicted in this img."
```

```
"Name the digit between 0 and 9 that this image represents."
"Which digit, 0 through 9, is displayed in this picture?"
"Tell which digit from 0 to 9 is shown in this image."
"Pinpoint the digit from 0-9 represented in this image."
"What digit from the range 0-9 is depicted in this image?"
"Indicate which digit (0-9) is illustrated in this image."
```

Given the question-answer pairs, either through querying GPT (SVG) or randomly sample from a pre-defined prompt pool, we use the following question template to construct the questions for our SGP-Bench (SVG):

```
Examine the following SVG code carefully and answer the
question based on your interpretation of the rendered image.

{SVG}

Question: {Question}
```

Given the question-answer pairs we use the following question template to construct the questions for our SGP-Bench (CAD):

```
Examine the following CAD code carefully to understand the
3D object it generates and answer the question based on your
interpretation of the rendered image of that object.

{CAD}

Hint: the CAD code has the following syntax:

{Hint}

Question: {Question}
```

When constructing the SGP-Bench (CAD), we also provide the syntax of CAD code:

```
CAD code consists of a sequence of CAD commands that describe
a 3D object.
The commands fall into two categories: sketch and extrusion.
Sketch commands are used to specify closed curves on a 2D plane
in 3D space. Each closed curve is referred as a loop, and one
or more loops form a closed region called a profile. A loop
always starts with an indicator command <SOL> followed by a
series of curve commands. All the curves on the loop are in
counterclockwise order, beginning with the curve whose starting
point is at the most bottom-left.
In total, there are three possible curve commands: Line, Arc,
and Circle.
Line(x, y): a line, with x, y as line end-point.
Arc(x, y, u, f): an arc, with x,y as arc end-point, u as sweep
angle and f as whether it is counter-clockwise, f=0 means it is
counter-clockwise, f=1 means it is not counter-clockwise.
Circle(x, y, r): a circle, with x,y as the center point and r
as the radius.
The extrusion command has two purposes:
1) It extrudes a sketch profile from a 2D plane into a 3D body,
and the extrusion type can be either one-sided, symmetric,
or two-sided with respect to the profile's sketch plane.
2) The command also specifies (through the parameter b in Ext)
how to merge the newly extruded 3D body with the previously
```

created shape by one of the boolean operations: either creating
a new body, or joining, cutting, or intersecting with the
existing body.
Ext(x, y, z, o, p, q, s, e, f, b, u): extrude operation, with
x, y, z as the sketch plane orientation, o, p, q as the sketch
plane origin, s as the scale of the associated sketch profile,
e, f as the extrude distances towards both sides, b as the type
of merge operation (could be New-body operation, join operation,
cut operation and intersect operation) and u as the extrude
type (could be one-sided, symmetric or two-sided).
<EOS> means the end of the code.

---

CAD code consists of a sequence of CAD commands that describe a
3D object.
The commands fall into two categories: sketch and extrusion.
Sketch commands are used to specify closed curves on a 2D plane
in 3D space. Each closed curve is referred as a loop, and one
or more loops form a closed region called a profile. A loop
always starts with an indicator command LOOP followed by a
series of curve commands.
Possible primitive types are defined with the following
parameters:
Arc(start_point,end_point,center_point,radius,normal,
start_angle,end_angle,reference_vector),
Circle(center_point,radius,normal), Line(start_point,
end_point), NurbsCurve(degree,knots,rational,
control_points,weights,periodic), Ellipse(major_axis,
major_axis_radius,minor_axis_radius,center_point,normal),
EllipticalArc(major_axis,major_axis_radius,
minor_axis_radius,center_point,normal).
The extrusion command ExtrudeFeature(operation, start_extent,
extent_type, extent_one, extent_two) has two purposes:
1) It extrudes a sketch profile from a 2D plane into a 3D body,
and the extrusion operation can be either one-sided, symmetric,
or two-sided with respect to the profile's sketch plane.
2) The command also specifies (extent_type) how to merge the
newly extruded 3D body with the previously created shape by one
of the boolean operations: either creating a new body, or
joining, cutting or intersecting with the existing body.

---

CAD code consists of a sequence of CAD commands that describe
a 2D object.
The commands fall into two categories: primitive and constraint
In total, there are five possible primitive types:
Point(x, y), Line(dirX, dirY, pntX, pntY, startParam, endParam)
Circle(xCenter, yCenter, xDir, yDir, radius, clockwise),
Arc(xCenter, yCenter, xDir, yDir, radius, clockwise, startParam
endParam), and Ellipse(xCenter, yCenter, xDir, yDir, radius,
minorRadius, clockwise).
x, y: the point coordinates.
dirX, dirY: the unit direction vector.
xCenter, yCenter: the coordinates the center point.
clockwise: a boolean value that indicates the orientation of
the unit direction vector.
pntX, pntY (Line): the coordinates of a point on the line.
startParam, endParam (Line): signed start/end point distances
to the point (pntX, pntY).
startParam, endParam (Arc): start/end angles to the unit

```
direction vector.
All primitives have an isConstruction boolean parameter
indicating if a primitive is to be physically realized or
simply serve as a reference for other primitives.
All constraints act on at least one primitive, indicated by the
corresponding number.
```

### F.2 TEMPLATE FOR EVALUATING MODELS ON SGP-BENCH

When evaluating different models on our SGP-Bench, we use the following evaluation template (multiple choice):

```
Answer the following multiple choice question. The last line of
your response should be of the following format:
'Answer: $LETTER' (without quotes) where LETTER is one of ABCD.
Think step by step before answering.

{Question}

A) {A}
B) {B}
C) {C}
D) {D}

Important, the last line of your response must be of the
following format:
'Answer: $LETTER' (without quotes) where LETTER must be one of
A, B, C or D.
```

When evaluating different models on our SGP-Bench, we use the following evaluation template (generation):

```
Solve the following problem step by step. The last line of your
response should be of the form Answer:
$ANSWER (without quotes) where $ANSWER is the answer to the
problem.

{Question}

Important, put your answer on its own line after "Answer:",
and you do not need to use a \\boxed command.
```

We use the following template, when we perform LLM-based evaluation:

```
Please read the following example. Then extract the answer
from the model response and type it at the end of the prompt.

Hint: The last line of your response should be of the following
format: 'Answer: $LETTER' (without quotes) where LETTER is
one of ABCD.
Question: What is the primary color of the object in the image?

A) Red
B) Blue
C) Black
D) Green

Model response: **Step 1: Examine the image**\n\nThe image
```

```
consists of various shapes filled with different colors.
We need to identify the primary color of the object in the
image.\n\n**Step 2: Focus on the dominant color**\n\nThe shapes
that cover the largest area in the object are filled with
shades of blue and its variations.\n\n**Answer: B**

Extracted answer: B

Hint: The last line of your response should be of the following
format: 'Answer: $LETTER' (without quotes) where LETTER is
one of ABCD.
Question: What is the background color of the image?

A) Red
B) Green
C) Blue
D) Yellow

Model response: Answer: The background color is blue.

Extracted answer: C

Hint: The last line of your response should be of the following
format: 'Answer: $LETTER' (without quotes) where LETTER is
one of ABCD.
Question: What is the shape of the buckle on the object?

A) Circle
B) Triangle
C) Square
D) Hexagon

Model response: Answer: D) Hexagon.

Extracted answer: D

Hint: The last line of your response should be of the following
format: 'Answer: $LETTER' (without quotes) where LETTER is
one of ABCD.
Question: What type of object is shown in the image?

A) Watch
B) Belt
C) Bracelet
D) Necklace

Model response: The object in the code is a watch.

Extracted answer: A

Hint: The last line of your response should be of the following
format: 'Answer: $LETTER' (without quotes) where LETTER is
one of ABCD.
Question: What is the primary color of the object in the image?

A) Blue
B) Yellow
C) Green
D) Red
```

```
Model response: The primary color of the object in the code
is yellow.

Extracted answer: B
```

