# OpenReview forum: "Can Large Language Models Understand Symbolic Graphics Programs?"
_ICLR.cc/2025/Conference — ICLR 2025 Spotlight_

### Official Review · Reviewer_rpKP · 2024-10-28

**Soundness:** 3
**Presentation:** 4
**Contribution:** 3
**Rating:** 8
**Confidence:** 3

**Summary:**

The authors explore the use of symbolic graphical programs (SGPs) as a means to assess the reasoning abilities of large language models (LLMs). SGPs are defined using visual markup languages and can be compiled by symbolic programs to produce images, such as SVGs in 2D or CAD models in 2D/3D. The central hypothesis is that the high-level reasoning skills of an LLM are linked to its capacity to (1) identify complex objects from primitive shapes and answer questions about them (Semantic Understanding) and (2) recognize when different sets of primitive shapes produce the same overall program (Consistency). To test this hypothesis quantitatively, the authors compiled approximately 4,000 program-question pairs for SVG images and 5,000 pairs for CAD programs. Their findings reveal that (1) open-weight models significantly lag behind proprietary ones in performance, (2) fine-tuning using an intermediate representation substantially improves reasoning performance, and (3) all models face challenges in distinguishing MNIST digits presented as SVG schemas.

**Strengths:**

- SGPs offer an intriguing middle ground between perception and reasoning. While humans can intuitively link rendered images to semantic concepts, associating these same concepts with the underlying SVG schema may demand additional time and coding knowledge.
- The dataset presented is highly scalable, as millions of SVG images and CAD programs with permissive licenses are available online for use in this benchmark.
- The proposed experiments are both well-motivated and effectively executed.
- The presentation is clear, consistent, and transparent.

**Weaknesses:**

* L108: "highly scalable benchmark." While I agree that SGPs have this potential, the paper does not demonstrate specific contributions toward achieving scalability. For example, benchmarks like [MineDojo](https://minedojo.org/) achieve high scalability by developing tools to efficiently scrape relevant data from platforms like YouTube, Reddit, and Wikia, resulting in a dataset of around 700k Minecraft videos and 6M+ comments. In contrast, the SGP dataset in this paper only contains ~5,000 data points. This is surprising given the abundance of online SVG images (for example, [SVGRepo](https://www.svgrepo.com/) claims to have over 500k SVGs). If SGP-Bench were a "highly scalable" benchmark, I would have expected an order of magnitude more data points or an explanation of the barriers preventing larger-scale sampling. Although this does not diminish the significance of the benchmark (Multimodal reasoning benchmark do not have to be "internet-scale" to be influential: https://arxiv.org/abs/2401.06209), the scalability claims should be adjusted accordingly
* The current definition of symbolic graphical programs might be overly broad. For instance, HTML could technically qualify as an SGP within the given definition since it uses a consistent markup language and rendering engines (like Chrome, Safari, or Firefox). While this isn’t inherently a weakness, it could impact the paper's novelty claims. If the SGP definition admits HTML as well, the authors might need to expand their related work and adjust their novelty claims to include efforts like https://arxiv.org/pdf/2210.03347, https://mbzuai-llm.github.io/webpage2code/, https://arxiv.org/abs/2403.09029 (and other relevant insights from work on action transformers for webpage understanding).

- [Figure 4] the comparison between OpenAI-o1 (a model likely using Chain of Thought reasoning) and standard zero-shot LLM queries using Llama3.1 70B seems problematic. However, this qualitative example showcases how  CoT reasoning might be a natural way for increasing LLM performance on this benchmark. Is there a particular reason the benchmarked models (Table 1)  do not evaluate the respective models under the CoT (or similar multi-step reasoning) paradigm?

- What are some potential "reasoning shortcuts" that could enable an LLM to perform well on this dataset without possessing true high-level reasoning abilities? For instance:
  - Data Leakage: Since the SVGs were sourced from the internet, it’s possible that proprietary LLMs were trained on some SVG/caption pairs. The authors propose the semantic consistency experiments to evaluate this. I believe that this partially mitigate this issue (as the jittered images likely wouldn't appear in the training set), but I'm concerned that jittering the image might not modify the underlying XML structure enough to conclusively reject the influence of data leakage. There are many ways to further validate this.
    - Calculate the tree edit distance (The Zhang Shasha algorithm is implemented in `pip install zss`) between the original and jittered SVG. Is the tree-edit-distance above a reasonable minimum threshold? Is the tree-edit-distance correlated with the relative performance difference? If low absolute correlation is observed, then the performance is invariant to the SVG structure, and data leakage might be absent.
    - Evaluate the performance of open-sourced models on obfuscated SVG files (Tools like https://github.com/svg/svgo will minify SVGs). If the minifier significantly alters the XML structure and performance remains unchanged, it could indicate no data leakage.
    - Of course, the authors may have other effective methods to test this. This isn’t necessarily a criticism of the paper, but rather a recognition that  the benchmark offers a unique opportunity to measure performance gains in LLM reasoning attributed to data leakage.

- L396: "No semantic instruction dataset directly over symbolic graphical programs." This paper https://mikewangwzhl.github.io/VDLM/ seems to propose an alternative semantically meaningful intermediate representation. I'd be interested in the authors' opinions on this paper.
- Generally, this paper is missing a discussion on the limitations of this benchmark. Here are some prompting questions:
  - Are all natural images expressible as SVGs? What does this mean for the comprehensiveness of this benchmark?
  -  Are SVGs / CAD programs representative of all types of symbolic graphical programs? What about visual specification languages like OpenGL, Taichi, etc? What about programming languages like Turtle/LOGO which admit recursive structures?
  - What are the limitations of SGP-Bench's scalability?

In conclusion, I'm currently recommending this paper for a **Weak Rejection**. While the benchmark is undoubtedly interesting to the ICLR community, it is crucial to reconcile the claims in this manuscript with those from the existing literature before the paper is accepted for publication. I am open to discussing these points with the authors and potentially revising my recommendation during the discussion period.

----

After discussions with the authors, I'm revising my recommendation to **Accept**. Overall, SGPs offer a promising avenue to assess the reasoning abilities of LLMs.

**Questions:**

(Merged with the Weaknesses Section)

---

> ### Author Response · Authors · 2024-11-20
> **Response to Reviewer rpKP (part 1)**
>
> We sincerely thank the reviewer for the constructive comments on our work. We take every comment seriously and hope our response can address the reviewer’s concerns. If there are any remaining questions, we are happy to address them. We summarize all the concerns into the following questions:
>
> ---
>
> **Q1: L108: "highly scalable benchmark." While I agree that SGPs have this potential, the paper does not demonstrate specific contributions toward achieving scalability. For example, benchmarks like MineDojo achieve high scalability by developing tools to efficiently scrape relevant data from platforms like YouTube, Reddit, and Wikia, resulting in a dataset of around 700k Minecraft videos and 6M+ comments. In contrast, the SGP dataset in this paper only contains ~5,000 data points. This is surprising given the abundance of online SVG images (for example, SVGRepo claims to have over 500k SVGs). If SGP-Bench were a "highly scalable" benchmark, I would have expected an order of magnitude more data points or an explanation of the barriers preventing larger-scale sampling. Although this does not diminish the significance of the benchmark (Multimodal reasoning benchmark do not have to be "internet-scale" to be influential: https://arxiv.org/abs/2401.06209), the scalability claims should be adjusted accordingly.**
>
> A1: Thank you for the great question! By 'highly scalable', we are referring to both the benchmark creation process and the creation of the instruction-tuning dataset. With this scalable pipeline, we are able to label more QA pairs for larger SVG datasets, but our benchmarking results show that the current scale already suffices. In addition, directly crawling internet-scale data can be costly for academia and demands more caution to copyright issues.
>
> Our benchmark comprises 6,740 unique questions (spanning SVG and CAD program codes), while our instruction-tuning dataset includes over 72,000 unique instruction-following pairs. During benchmark creation, we used our pipeline to generate 25,167 questions for SVG programs and 78,198 questions for CAD programs. Specifically, for the 3D subset, we selected 1,000 questions from an initial set of 32,307; for the 2D subset, 700 questions from 20,455; and for the 3D_complex subset, 700 questions from 25,536. To ensure variety and quality, we manually reviewed the generated questions to remove any that were repetitive or too visually similar, finalizing a subset of 6,740 questions for our SGP-Benchmark.
> Our evaluation of current LLMs indicates that this benchmark is sufficiently large and diverse to highlight performance differences across different models.
>
> Thanks for mentioning this paper (https://arxiv.org/abs/2401.06209). We agree with the reviewer that we can adjust our scalability claim to make it more justified. We will make a more detailed discussion of our scalability claim and our benchmark scale in the revised paper.
>
> ---
>
> **Q2: The current definition of symbolic graphical programs might be overly broad. For instance, HTML could technically qualify as an SGP within the given definition since it uses a consistent markup language and rendering engines (like Chrome, Safari, or Firefox). While this isn’t inherently a weakness, it could impact the paper's novelty claims. If the SGP definition admits HTML as well, the authors might need to expand their related work and adjust their novelty claims.**
>
> A2: We appreciate the reviewer's suggestion. However, we would like to clarify that our work does not involve HTML, as our focus is on assessing LLMs' understanding of 'symbolic graphics programs.' Following the definition of 'symbolic graphics programs' from [1], we specifically focus on symbolic programs that uniquely produce visual data. Such representations serve as compact, interpretable, high-quality formats for graphics content. Beyond SVG and CAD, other symbolic graphics programs that fit this definition include constructive solid geometry (CSG), boundary representations (B-reps), L-systems, and shape grammars, with SVG and CAD representing the most prominent examples. Other symbolic graphics programs can easily adopt our pipeline to create additional benchmark tasks.
>
> Our SGP benchmark encompasses 2D primitive shapes, 3D geometry, and complex 2D renderings, facilitating both zero-shot testing with standard, known syntax (SVG) and in-context learning with domain-specific languages (DSL). This range substantiates our claim that our benchmark is able to assess LLM’s understanding of symbolic graphical programs.
> We will add HTML-related studies to our related work section and further clarify the distinctions between our approach and HTML-based methods.
>
> [1] Ritchie, Daniel, et al. "Neurosymbolic models for computer graphics." Computer graphics forum. Vol. 42. No. 2. 2023.

---

> > ### Author Response · Authors · 2024-11-20
> > **Response to Reviewer rpKP (part 2)**
> >
> > **Q3: [Figure 4] the comparison between OpenAI-o1 (a model likely using Chain of Thought reasoning) and standard zero-shot LLM queries using Llama3.1 70B seems problematic. However, this qualitative example showcases how CoT reasoning might be a natural way for increasing LLM performance on this benchmark. Is there a particular reason the benchmarked models (Table 1) do not evaluate the respective models under the CoT (or similar multi-step reasoning) paradigm?**
> >
> > A3: Thank you for the question. To clarify, we would like to emphasize that all the models evaluated in this paper (including Table 1) already use standard zero-shot CoT prompting. We will make this point clearer in our revised paper to avoid any potential misunderstanding.
> >
> > Our main purpose in presenting the qualitative reasoning steps of OpenAI-o1 and LLaMA is to illustrate how the tested LLMs tackle tasks in our benchmark through a range of sophisticated reasoning skills, including numeric perception, spatial reasoning, geometric understanding, long-range planning, and semantic common sense reasoning. The qualitative examples in the paper illustrate how OpenAI-o1, with its more advanced test-time reasoning capabilities, achieves a more accurate conclusion on the challenging SVG-MNIST tasks.
> >
> > We agree with the reviewer that when employing more sophisticated test-time scaling, the tested models should have better performance in our benchmark. This observation further highlights our benchmark as a valuable tool for evaluating future test-time scaling methods.

---

> > > ### Author Response · Authors · 2024-11-20
> > > **Response to Reviewer rpKP (part 3)**
> > >
> > > **Q4: What are some potential "reasoning shortcuts" that could enable an LLM to perform well on this dataset without possessing true high-level reasoning abilities? For instance: Data Leakage: Since the SVGs were sourced from the internet, it’s possible that proprietary LLMs were trained on some SVG/caption pairs. The authors propose the semantic consistency experiments to evaluate this. I believe that this partially mitigate this issue (as the jittered images likely wouldn't appear in the training set), but I'm concerned that jittering the image might not modify the underlying XML structure enough to conclusively reject the influence of data leakage. There are many ways to further validate this. Calculate the tree edit distance (The Zhang Shasha algorithm is implemented in pip install zss) between the original and jittered SVG. Is the tree-edit-distance above a reasonable minimum threshold? Is the tree-edit-distance correlated with the relative performance difference? If low absolute correlation is observed, then the performance is invariant to the SVG structure, and data leakage might be absent. Evaluate the performance of open-sourced models on obfuscated SVG files (Tools like https://github.com/svg/svgo will minify SVGs). If the minifier significantly alters the XML structure and performance remains unchanged, it could indicate no data leakage.**
> > >
> > > A4: Thank you for the suggestion! The average tree edit distance (calculated using the Zhang-Shasha algorithm and based on the implementation referred by the reviewer) for our invariance test per SVG program is **11.9**, ranging from 2 to 43. On average, our code-level perturbations require 12 node edit operations to revert back to the original SVG code, indicating significant changes in both numerical values and code structure.
> > > To assess any correlation between tree edit distance and consistency results, we sampled 100 questions with a tree edit distance between 5–10 and 100 questions with a tree edit distance between 20–25 from our consistency dataset. The consistency results for Llama3.1-70B are presented in the table below. We conclude that varying levels of SVG code structure modifications do not affect performance. This suggests that the performance of LLMs is based on actual code reasoning, rather than the familiarity of the code.
> > >
> > > | Tree Edit Distance | Trans. Consistency | Rot. Consistency |
> > > |------------|---------------------|------------------|
> > > | 5-10       | 0.890              | 0.833           |
> > > | 20-25      | 0.882              | 0.822           |
> > >
> > >
> > > We appreciate the reviewer’s suggestion and have utilized the SVGO tool to minify the SVG images in our main benchmark, subsequently evaluating models on these minified SVG program codes while keeping everything else (including the question prompt and multiple-choice options) unchanged. The results are presented in the table below. As shown, there is no notable change in the tested LLM’s performance, supporting our hypothesis that data leakage has not influenced model performance. More importantly, the results also suggest that the performance is based on actual reasoning of the symbolic programs, because removing the function-irrelevant program components does not affect the performance. We find this testing approach suggested by the reviewer particularly valuable for reinforcing our benchmark’s robustness, and plan to include additional results on these minified SVG programs in the revised paper.
> > >
> > > | Model        | Without Minifier (paper result) | With Minifier |
> > > |---------------|------------------------------------------|------------------|
> > > | Mistral-7B-v0.3 | 0.417                                 | 0.414            |
> > > | LLama3.1-8B   | 0.465                                 | 0.465            |
> > > | LLama3.1-70B | 0.574                                 | 0.578            |
> > > | Qwen2-72B | 0.537                                     | 0.538            |
> > >
> > > We want to emphasize that our findings contrast with recent studies suggesting that LLMs’ reasoning is largely probabilistic pattern-matching rather than formal reasoning, where small input variations can drastically alter outputs, reducing performance on established benchmarks (e.g., GSM8K [1][2]). While we refrain from commenting on these hypotheses, our benchmark offers a way to test generic reasoning abilities of LLMs that are invariant to large modifications of the input. Our experiments have demonstrated that both numerical and structural changes in the program code do not affect the LLM’s ability to perform on our benchmark, showing that our benchmark is able to robustly evaluate LLM’s intrinsic reasoning abilities.
> > >
> > > [1] Mirzadeh, Iman, et al. "Gsm-symbolic: Understanding the limitations of mathematical reasoning in large language models." arXiv preprint arXiv:2410.05229 (2024).
> > >
> > > [2] Jiang, Bowen, et al. "A Peek into Token Bias: Large Language Models Are Not Yet Genuine Reasoners." arXiv preprint arXiv:2406.11050 (2024).

---

> > > > ### Author Response · Authors · 2024-11-20
> > > > **Response to Reviewer rpKP (part 4)**
> > > >
> > > > **Q5: L396: "No semantic instruction dataset directly over symbolic graphical programs." This paper https://mikewangwzhl.github.io/VDLM/ seems to propose an alternative semantically meaningful intermediate representation. I'd be interested in the authors' opinions on this paper.**
> > > >
> > > > A5: We thank the reviewer for mentioning this work VDLM (https://arxiv.org/abs/2404.06479), as it has internal connections with our benchmark.
> > > > We appreciate the authors' efforts to enhance large multimodal models (LMMs) and their introduction of the Primal Visual Description (PVD), a novel intermediate textual representation designed to simplify reasoning about vector graphics.
> > > >
> > > > Our work differs primarily in focus: we probe LLMs' inherent ability to engage in visual imagination based on local curvatures and strokes, whereas the VDLM authors aim to improve LMMs’ capability by introducing a 'preprocessing' step that provides additional information to the LLMs.
> > > >
> > > > We recognize the authors' emphasis on enhancing low-level visual perception—particularly in tasks centered on shapes and sizes—as LMMs continue to face challenges with visual grounding. While we agree that improved low-level visual perception benefits high-level visual reasoning, we found that transforming visual input into SVG and then into an intermediate, caption-like representation can be limiting. For example, when dealing with complex natural images rather than simple images with primitive shapes (as shown in Figure 3 of the VDLM paper), the resulting SVG code generated by VTracer becomes extremely long, making it challenging to extract meaningful information for PVD.
> > > >
> > > > Our findings on LLMs' visual imagination abilities align with the VDLM authors' observations (see Table 1 in VDLM). Specifically, VDLM-txt, even without access to the original image, outperforms strong LMMs. While the VDLM authors attribute this to their PVD representation, our results demonstrate that this ability scales with model size in LLMs, allowing complex spatial reasoning based solely on program code, without the need of an intermediate representation.
> > > >
> > > > ---
> > > >
> > > > **Q6: Generally, this paper is missing a discussion on the limitations of this benchmark. Here are some prompting questions:
> > > > Are all natural images expressible as SVGs? What does this mean for the comprehensiveness of this benchmark?
> > > > Are SVGs / CAD programs representative of all types of symbolic graphical programs? What about visual specification languages like OpenGL, Taichi, etc? What about programming languages like Turtle/LOGO which admit recursive structures?
> > > > What are the limitations of SGP-Bench's scalability?**
> > > >
> > > > A6: Yes, we agree with the reviewer that we should add a section to discuss in depth the limitation of the benchmark. Currently, our discussion of limitations mostly focuses on the capability of LLMs (see section 7), but we appreciate the incentives that the reviewer provides to discuss the limitations of our benchmark. We will add a separate limitation section in our paper.
> > > >
> > > > Our paper starts with two prominent symbolic graphics programs: SVG and CAD, with the potential to easily extend to other symbolic graphics programs such as Turtle and LOGO. However, our benchmark creation does require that the symbolic graphics programs should generate semantically meaningful visual content such that they can be used to prompt large vision-language models to have meaningful semantic questions. This is also the reason we choose to use more popular symbolic graphics programs such as SVG and CAD.

---

> > > > > ### Comment · Reviewer_rpKP · 2024-11-21
> > > > >
> > > > > Thank you for the responses. Most of my concerns have also been addressed. As such, I'm revising my recommendation towards acceptance. Many thanks for the additional experiments and the great work!

---

> > > > > > ### Author Response · Authors · 2024-11-21
> > > > > > **Thanks for your time!**
> > > > > >
> > > > > > We appreciate the reviewer for spending the time to read our long rebuttal. We are super glad that the reviewer's concerns are now addressed.

---

### Official Review · Reviewer_ESu8 · 2024-10-30

**Soundness:** 3
**Presentation:** 2
**Contribution:** 4
**Rating:** 8
**Confidence:** 5

**Summary:**

This paper introduces a benchmark designed to assess LLMs'  comprehension of symbolic graphics programs in SVG format, two 3D CAD formats (DeepCAD and Fusion360), and one 2D CAD format (SketchGraphs). The authors present evaluation results across several LLMs and introduce a symbolic instruction dataset, showing that fine-tuning on this dataset can enhance LLM reasoning capabilities in general.

**Strengths:**

1. The benchmark dataset is large, with 4,340 questions in the SVG set and 2,400 in the CAD set.
2. Both 2D and 3D objects are evaluated.
3. The paper addresses data leakage issues by utilizing global transformations like translations and rotations.
4. The proposed instruction tuning dataset improves general reasoning ability in LLMs.
5. The paper provides a critical view on current LLMs' capability.

**Weaknesses:**

1. The human study reports 90% labeling accuracy, leaving 10% mislabeled data, which could significantly impact the results given limited variation of the accuracy in LLM's evaluation. I suggest doing a human filtering across the whole benchmark to improve the accuracy of the ground truth answer.
2. The method for generating questions on 3D dataset from limited rendered views may lead to errors, especially in counting tasks; for example, the second part Figure 7 (row 3, column 3) is confusing. The question "How many visible cylindrical pins are there on the CAD object?" doesn't make sense since "visible" doesn't have a meaning given a code representing a 3D object without perspective information.
3. Several parts of the writing need further clarification (see questions below).

**Questions:**

1. In L76-L77, the authors claim that "The order of symbolic operations may substantially affect semantic meaning". However, this does not seem applicable to SVG, where objects are defined by coordinates and unaffected by drawing order. Could this be clarified?
2. In Figure 6, what qualifies as an "operation" in vector graphics? For example, does a polyline with multiple vertices count as one operation or several?
3. Inconsistencies in notation between Figure 6(b) (symbol $3D_{Recon}$) and the text (symbol $3D_{complex}$ in L257-L258) need clarification.
4. During fine-tuning, was there any overlap between training and test datasets? Additionally, could more detail be provided on the construction of the symbolic instruction dataset such as whether it includes both SVG and CAD formats?
5. If the enhanced answer extraction with LLM is applied consistently across all experiments or is only applied to models trained on SIT data?

---

> ### Author Response · Authors · 2024-11-20
> **Response to Reviewer ESu8 (1/2)**
>
> We sincerely thank the reviewer for the constructive comments on our work. We take every comment seriously and hope our response can address the reviewer’s concerns. If there are any remaining questions, we are happy to address them. We summarize all the concerns into the following questions:
>
> ---
>
> **Q1: In L76-L77, the authors claim that "The order of symbolic operations may substantially affect semantic meaning". However, this does not seem applicable to SVG, where objects are defined by coordinates and unaffected by drawing order. Could this be clarified?.**
>
> A1: We appreciate the reviewer's observation that, in SVG, the order of symbolic operations does not always affect semantic meaning. However, in many cases, the operation order is crucial for the final rendering or shape. For instance, in SVG, elements defined earlier are drawn first, with subsequent elements layered on top. Changing the order of elements alters their stacking order on the canvas, which is especially evident with overlapping shapes, where later elements may cover or partially obscure earlier ones. Similarly, in CAD program codes, altering the order of operations can significantly impact the final shape. For example, an 'extrude' operation is defined based on a preceding 'profile' operation, transforming a 2D sketch into a 3D shape.
>
> ---
>
> **Q2: In Figure 6, what qualifies as an "operation" in vector graphics? For example, does a polyline with multiple vertices count as one operation or several?**
>
> A2: For our analysis, we count the number of drawing-related operations that contribute to the visual output, including commands such as 'circle,' 'rect,' 'line,' 'polyline,' 'polygon,' 'path,' and similar. Therefore, each 'polyline' operation with multiple vertices is counted as a single operation.
>
> ---
>
> **Q3: Inconsistencies in notation between Figure 6(b) and the text (in L257-L258) need clarification.**
>
> A3: Great catch! We apologize for the mistake we made in the notations, the correct notation should be as the text 3D_complex. We will update the figure in the revised paper.
>
> ---
>
> **Q4: During fine-tuning, was there any overlap between training and test datasets? Additionally, could more detail be provided on the construction of the symbolic instruction dataset such as whether it includes both SVG and CAD formats?**
>
> A4: Thank you for raising this concern. We carefully apply a filtering step to ensure that no data from the test set is included in the fine-tuning dataset.
> We are happy to provide additional details regarding the construction of the SIT dataset. Currently, our proposed SIT dataset contains only SVG data. The primary purpose of SIT is to offer insights to the research community on how SIT can enhance general reasoning ability. Prior research has shown that training on general code data can improve reasoning skills [1,2]. For example, several code-focused LLMs [3,4,5] have demonstrated enhanced reasoning abilities in solving mathematics problems by incorporating more code data [6]. Symbolic graphics programs, a specialized form of code data, are used to generate 2D and 3D graphics content. They share a hierarchical structure similar to that of general code data, but unlike traditional programs that typically produce numerical outputs, symbolic graphics programs generate outputs rich in semantic information, involving complex reasoning tasks such as part/component localization, color identification, affordance prediction, semantic understanding, and geometric reasoning. For instance, consider a question about a symbolic graphics program: 'What is the color of the top part of the object?' Answering this requires the LLM to identify which part of the code corresponds to the top section and determine the color based on the color code. Similarly, a question like 'What is the object primarily used for?' demands that the LLM interpret the object’s semantic meaning and suggest a suitable application. These questions involve multiple, interdependent reasoning steps, where an error at any step would result in an incorrect final answer.
> In future work, the same pipeline could be applied to create instruction-following datasets for other types of symbolic graphics programs. More importantly, the data creation pipeline is quite scalable and requires minimal human interference.
>
> [1] At Which Training Stage Does Code Data Help LLMs Reasoning? ICLR 2024
>
> [2] To Code, or Not To Code? Exploring Impact of Code in Pre-training, https://arxiv.org/abs/2408.10914
>
> [3] Code Llama: Open Foundation Models for Code, https://arxiv.org/abs/2308.12950
>
> [4] DeepSeek-Coder: When the Large Language Model Meets Programming -- The Rise of Code Intelligence, https://arxiv.org/abs/2401.14196
>
> [5] WizardCoder: Empowering Code Large Language Models with Evol-Instruct, https://arxiv.org/abs/2306.08568
>
> [6] DeepSeekMath: Pushing the Limits of Mathematical Reasoning in Open Language Models, https://arxiv.org/abs/2402.03300

---

> > ### Author Response · Authors · 2024-11-20
> > **Response to Reviewer ESu8 (2/2)**
> >
> > **Q5: If the enhanced answer extraction with LLM is applied consistently across all experiments or is only applied to models trained on SIT data?**
> >
> > A5: LLM-based answer extraction is applied only to models trained with SIT data. This is because, after fine-tuning with SIT data, we observe distinct response patterns across models. For example, in our experience, models from the Mistral LLM series tend to be more cautious and may occasionally refuse to answer when uncertain, or they may not strictly adhere to the question prompt format.
> > In our main benchmark (Table 1 in the main paper), we use a matching-based answer extraction method for all tested models to ensure a fair comparison. Moreover, the models tested in Table 1 have not been finetuned with SIT data and possess good instruction-following ability. This approach is also very common in other benchmarks, such as MathVista. In the instruction-tuning experiment, we focus on assessing the models' understanding of graphics programs while controlling for other variables, which is why we use LLM-based evaluation.
> > We also want to note that in Table 3 of the main paper, where we report model performance on various popular LLM evaluation benchmarks after training on SIT data, we always follow the original evaluation protocol of each corresponding benchmark to ensure that the results we obtained are valid.
> >
> > ---
> >
> > **Q6: The human study reports 90% labeling accuracy, leaving 10% mislabeled data, which could significantly impact the results given limited variation of the accuracy in LLM's evaluation. I suggest doing a human filtering across the whole benchmark to improve the accuracy of the ground truth answer.**
> >
> > A6: Thanks for the comment and the suggestion! We indeed manually filter out some invalid questions. First of all, the 90% human mode accuracy does not imply that these questions are mislabeled, and it is more of the case that these benchmark questions are rather difficult and require some domain knowledge. Specifically, our question filtering is done by multiple domain experts in 3D vision, while the human mode accuracy is computed by 5 random annotators. Therefore, there might still be some mismatch between experts and ordinary annotators. However, we have double-checked the benchmark questions and ensure that they have the correct ground truth answers.
> >
> > ---
> >
> > **Q7: The method for generating questions on 3D dataset from limited rendered views may lead to errors, especially in counting tasks; for example, the second part Figure 7 (row 3, column 3) is confusing. The question "How many visible cylindrical pins are there on the CAD object?" doesn't make sense since "visible" doesn't have a meaning given a code representing a 3D object without perspective information.**
> >
> > A7: We appreciate the reviewer’s thorough inspection of the provided benchmark examples. As our benchmark creation pipeline depends on the visual understanding of LLMs, and given that current multimodal LLMs still face challenges with certain visual tasks—including counting, as the reviewer noted (see https://arxiv.org/pdf/2401.06209)—some limitations remain. Although this pipeline allows for the creation of benchmark questions without human effort, we still need to manually inspect the generated questions to ensure the correctness of ground truth labels. Initially, our benchmark consisted of 25,167 questions on SVG programs and 78,198 questions on CAD programs, from which we manually selected those with correct answers to include into our SGP-Benchmark.
> > During manual inspection, we found that LLMs answered most questions correctly, though some responses were incorrect, occasionally due to ambiguous phrasing. We removed questions with incorrect answers to finalize our SGP benchmark.
> > For the particular question the reviewer mentioned: “How many visible cylindrical pins are there on the CAD object?”, it is actually a valid question. The “visibility” here means whether the component of the 3D object is visible in terms of the general 3D space (agnostic to the perspective information), as “invisible” means that a component is warped inside other components. However, we do agree with the reviewer that some questions might cause some confusion, but this is only a very small fraction of our benchmark questions.
> > Alternatively, one could also use the unfiltered questions to assess an LLM’s general ability to visually imagine based on low-level code, based on how well an LLM aligns with a VLM using visual input, as each program code uniquely corresponds to a specific 2D or 3D content.

---

> ### Comment · Reviewer_ESu8 · 2024-11-20
>
> Thank you for your response. Most of my concerns are addressed but I'm still confused on few points.
> 1. You said that "Initially our benchmark consisted of 25,167 questions on SVG programs and 78,198 questions on CAD programs", but in your paper L215 and L255 you said the final benchmark contains 4,340 questions in the SVG set and 2,400 questions in the CAD set. In your human filtering process, are you only selecting 17% of your generated SVG questions and 3% of your generated CAD questions? The passing rate looks inconsistent with the passing rate of 90% in the human checking process.
> 2. I'm not convinced that "models tested in Table 1 have not been finetuned with SIT data and possess good instruction-following ability". Some models in table 1 are small and don't have good instruction-following ability. Therefore, I think it might be the enhanced answer extraction with LLM improving the performance (e.g. by making the output format consistent and preventing errors due to format). Could you please report at least one result of those small models (without SIT) using enhanced answer extraction with LLM to make the experiment fair? It don't need to be large scale, only a small portion of data proving that enhanced answer extraction won't affect accuracy is fine.

---

> > ### Author Response · Authors · 2024-11-20
> > **Second round response to Reviewer ESu8**
> >
> > We are glad that our previous responses are able to address most of your concerns and are delighted to provide some more details to the follow-up questions. If there is any remaining concern, we are more than happy to address them.
> >
> > ---
> >
> > **Q1:You said that "Initially our benchmark consisted of 25,167 questions on SVG programs and 78,198 questions on CAD programs", but in your paper L215 and L255 you said the final benchmark contains 4,340 questions in the SVG set and 2,400 questions in the CAD set. In your human filtering process, are you only selecting 17% of your generated SVG questions and 3% of your generated CAD questions? The passing rate looks inconsistent with the passing rate of 90% in the human checking process..**
> >
> > A1: Thanks for the question. The reason behind the low passing rates is in fact our design choice, rather than the validity of the questions. In fact, most questions are valid for evaluating LLMs, but we filter out some questions based on the code length, question type, code style and the diversity of content structure. We aim to construct a benchmark that is comprehensive yet informative. In contrast, our human evaluation study aims to compare the prediction entropy, instead of checking the validity of questions.
> >
> > Our objective in designing this benchmark was to strike an optimal balance: the number of tasks should be substantial enough to discernibly differentiate between the capabilities of various large language models (LLMs), yet not so burdensome in computational load and time that they become impractical for large-scale testing. The size of our final benchmark aligns with that of other well-established LLM benchmarks, such as MathVista. The current testing time of Llama3.1-70B on the SVG set is already around 70 minutes with 8 NVIDIA-H100-80GB GPUs. Therefore, we aim to limit the number of total questions to reduce the evalutation cost.
> >
> > Another benchmark design choice was to filter out questions with excessively long program code. Given that symbolic program code forms a part of the input question prompt, we needed to exclude many lengthy codes to ensure that the benchmark remains compatible with most LLMs, including older generations (due to the context window size). Moreover, we avoid retaining only the shortest codes to prevent the benchmark from being too simple, as this would limit the complexity of the resulting graphical content.
> >
> > Furthermore, we also filter out repetitive or overly simple questions. This explains why there is a 90% human mode accuracy rate among random annotators, even after meticulous manual review. Our goal is to maintain questions that present a certain level of complexity. Additionally, we drop some symbolic graphics programs that yield too similar renderings or 3D shapes, enhancing the diversity and quality of our benchmark.
> >
> > As a result, we have constructed a high-quality, comprehensive benchmark that covers a wide range of semantic graphics program understanding tasks. This includes dealing with 2D primitive shapes, 3D geometry, and complex 2D renderings. It supports zero-shot testing using standard, popular syntax (SVG) and in-context learning through domain-specific languages (DSL), eg. CAD. Importantly, this benchmark is structured to be manageable in terms of computational demands, thus making it more accessible to the broader research community.
> >
> > ---
> >
> > **Q2: I'm not convinced that "models tested in Table 1 have not been finetuned with SIT data and possess good instruction-following ability". Some models in table 1 are small and don't have good instruction-following ability. Therefore, I think it might be the enhanced answer extraction with LLM improving the performance (e.g. by making the output format consistent and preventing errors due to format). Could you please report at least one result of those small models (without SIT) using enhanced answer extraction with LLM to make the experiment fair? It don't need to be large scale, only a small portion of data proving that enhanced answer extraction won't affect accuracy is fine..**
> >
> > A2: Thanks for the question. First, we would like to emphasize that matching-based answer extraction is a valid and fair evaluation approach, as the ability to follow instructions is an integral part of an LLM's capabilities. However, to address the reviewer’s concern, we spend some time to conduct the following experiment. Using Gemma-1.1-2B, the LLM with the fewest parameters in our study, we compare the results of matching-based evaluation with LLM-based evaluation. Our findings show that LLM-based evaluation has only a very limited impact on performance. For consistency with the number of papers analyzed, we used the full dataset when performing the LLM-based evaluation.
> >
> > |                                | SVG   | CAD   |
> > |--------------------------------|-------|-------|
> > | Gemma-1.1-2B (matching-based)  | 0.317 | 0.278 |
> > | Gemma-1.1-2B (llm-based)       | 0.320 | 0.280 |

---

> > > ### Comment · Reviewer_ESu8 · 2024-11-20
> > >
> > > Thank you for your response. All my concerns are addressed, therefore I increased my ratings (5->8).

---

> > > > ### Author Response · Authors · 2024-11-21
> > > > **Thansk for the response!**
> > > >
> > > > We appreciate the reviewer's efforts for reading our rebuttal and are very glad to see the reviewer's concerns are addressed now.

---

### Official Review · Reviewer_zWHc · 2024-10-31

**Soundness:** 3
**Presentation:** 3
**Contribution:** 3
**Rating:** 6
**Confidence:** 4

**Summary:**

The paper assesses large language models (LLMs) for their ability to "understand" raw symbolic graphics programs like SVG and CAD without direct visual input. By creating a benchmark (SGP-Bench), the authors test this through tasks requiring semantic-level answers and transformation invariance, providing insight into how LLMs interpret non-image-based graphics data. The authors also introduce Symbolic Instruction Tuning (SIT), a method to fine-tune LLMs with specific symbolic graphics data, show that its enhancing both specialized and general reasoning abilities.

**Strengths:**

1. Innovative in presenting a benchmark and dataset that adds value to multi-modal LLM and vision foundation model research.
2. Extensive experimentation with well-designed elements, particularly in the fine-grained categorization of SVG-Understanding. While categories like "color" and "shape" might be addressed through subpattern matching in SVG codes, "Semantics" and "Reasoning" better assess true reasoning capability.
3. Clear presentation and valuable insights.

**Weaknesses:**

1. The SGP-MNIST experiments show limited spatial reasoning in LLMs, suggesting that some claims about models’ spatial reasoning or “visual imagery” abilities may be overstated.
2. The conclusion is somewhat underwhelming, reiterating known principles such as scaling laws and fine-tuning effects.

**Questions:**

1. Regarding Weakness 1, it appears the model learns from structured data without evidence of internal image rendering, likely due to lack of instruction. It would be interesting to test whether explicitly prompting for “internal rendering” as a Chain-of-Thought (CoT) step impacts performance.
2. I'm curious whether any experiments have been conducted using rendered images as input in multi-modal LLMs to compare performance with raw SVG code inputs, providing insight into how well models perform with visual representations.
3. In Appendix E.2, it’s noted that fine-tuning SIT-55k on Llama3-8B actually degrades performance. Is this due to answers not following the specified template, or are the answers correctly formatted but simply inaccurate?

---

> ### Author Response · Authors · 2024-11-20
> **Response to Reviewer zWHc (1/2)**
>
> We sincerely thank the reviewer for the constructive comments on our work. We take every comment seriously and hope our response can address the reviewer’s concerns. If there are any remaining questions, we are happy to address them. We summarize all the concerns into the following questions:
>
> ---
>
> **Q1: Regarding Weakness 1, it appears the model learns from structured data without evidence of internal image rendering, likely due to lack of instruction. It would be interesting to test whether explicitly prompting for “internal rendering” as a Chain-of-Thought (CoT) step impacts performance.**
>
> A1: Thank you for the question! In fact, our question prompt does explicitly use Chain-of-Thought (CoT) reasoning, and our prompt also implicitly includes the concept of 'internal rendering' as part of the prompt itself.
>
> Below we give our original prompt templates:
>
> ```
> Answer the following multiple choice question. The last line of your response should be of the following format: 'Answer: $LETTER' (without quotes) where LETTER is one of ABCD. Think step by step before answering.
>
> [**Question prompt**]
>
> A) [A]
> B) [B]
> C) [C]
> D) [D]
> ```
>
> The [**Question prompt**] consists of the following instructions and the corresponding symbolic program:
>
> ```
> Examine the following CAD code carefully to understand the 3D object it generates and answer the question based on your interpretation of the rendered image of that object.
> [**CAD program code**] + [**Benchmark question**]
> ```
>
> ```
> Examine the following SVG code carefully and answer the question based on your interpretation of the rendered image.
> [**SVG program code**] + [**Benchmark question**]
> ```
>
> To address your question, we have included an additional test that explicitly uses the prompt for 'internal rendering':
>
> | Model        | Without Explicit Mention of Internal Rendering | With Explicit Mention of Internal Rendering |
> |--------------|-----------------------------------------------|--------------------------------------------|
> | LLama3.1-70B | 0.574                                         | 0.579                                      |
>
> As shown in the table above, explicitly incorporating 'internal rendering' as part of the CoT slightly enhances model performance. This finding suggests that the ability of internal rendering is beneficial for solving the tasks in our benchmark. Moreover, the improvement is marginal, implying that the “visual imagery” ability is fundamental, and simply using better instruction prompts can not easily improve it.
>
> ---
>
> **Q2: I'm curious whether any experiments have been conducted using rendered images as input in multi-modal LLMs to compare performance with raw SVG code inputs, providing insight into how well models perform with visual representations.**
>
> A2: Thanks for the question! Our SGP benchmark is automatically constructed by leveraging the visual understanding capabilities of LLMs, meaning that the questions are designed to be easily answerable from visual input (see Figure 7 in our main paper). In our benchmark, GPT-4’s responses to questions using rendered images serve as the ground truth labels. To ensure both accuracy and diversity in our benchmark, we manually reviewed the generated question-answer pairs and removed the questions with incorrect answers. As a result, GPT-4 with visual input achieves nearly perfect accuracy on the benchmark questions.
>
> ---
>
> **Q3: In Appendix E.2, it’s noted that fine-tuning SIT-55k on Llama3-8B actually degrades performance. Is this due to answers not following the specified template, or are the answers correctly formatted but simply inaccurate?**
>
> A3: Thanks for the question! We appreciate the reviewer for taking the time to examine the experiments in our Appendix. For evaluation, we used an LLM-based approach to assess the responses of the SIT-finetuned models, instead of matching-based approaches. There are almost no template mismatches under this evaluation protocol. The degradation of performance is indeed caused by the incorrect answers.
>
> ---
>
> **Q4: The SGP-MNIST experiments show limited spatial reasoning in LLMs, suggesting that some claims about models’ spatial reasoning or “visual imagery” abilities may be overstated.**
>
> A4: Thanks for the comment. Most existing LLMs indeed perform poorly on the SGP-MNIST benchmark. This experiment aims to demonstrate the task of symbolic program understanding can be extremely challenging and is far from being solved at the moment. However, from Figure 4 in the main paper, we can observe that stronger LLMs (e.g. OpenAI-o1) can perform qualitatively better than less powerful LLMs (e.g. Llama-3.1). Such results suggest that the ability of symbolic program understanding/reasoning is fundamental and it can serve as a good indicator of LLM’s reasoning abilities. More importantly, the current results on our SGP-MNIST benchmark shows that there is still much room for existing LLMs to improve.

---

> > ### Author Response · Authors · 2024-11-20
> > **Response to Reviewer zWHc (2/2)**
> >
> > **Q5: The conclusion is somewhat underwhelming, reiterating known principles such as scaling laws and fine-tuning effects.**
> >
> > A5: Thanks for the comment. We want to emphasize our contributions are quite unique in the following aspects:
> >
> > - **What motivates symbolic graphics program understanding**
> >
> > Symbolic graphics programs are a structured representation of the visual world. Due to the built-in support of code, it is a great representation for LLMs to perceive the visual world, especially compared to current vision-language LLMs where the vision-language bridge is implemented by a cross-modal encoder (eg, CLIP). Therefore, we aim to find out to what extent LLMs can understand symbolic graphics programs.
> >
> > - **Significance of symbolic graphics program understanding**
> >
> > Symbolic graphics programs, to a certain degree, can be seen as an intermediate abstraction between pixel-valued images and pure language descriptions of them: straight lines are now expressed as a single line of code instead of dozens of pixels, non-local augmentations can be done by altering colors and parameters, etc. It is natural to explore to what abstraction level LLMs can understand images. SGP-Bench is one of the first benchmarks to evaluate symbolic graphics program understanding.
> >
> > Unlike a typical numerical output of generic programs, symbolic graphics programs uniquely output semantic objects (eg, images, 3D geometry) that can be understood in many different ways and subsumes many reasoning questions (eg, localization, identification, affordance prediction). SIT involves a multi-facet distillation of both vision encoders and LLMs
> >
> > Moreover, we believe that understanding such a representation is a fundamental skill of “imagination without images”, on which being proficient implicitly helps other related tasks.
> >
> > For application, because symbolic graphics programs are a structured representation of the visual world, a natural application is to consider a symbolic “LLaVa”, where the vision encoder is no longer needed and LLMs can perceive the visual world via symbolic graphics programs.
> >
> > - **Symbolic instruction tuning is novel as it distills visual knowledge to LLMs**
> >
> > The proposed SIT method can generally improve reasoning abilities of LLMs, as verified in our experiments. We want to highlight that SIT distills knowledge from the visual world (with image-text correspondence). This is because the detailed description of the symbolic graphics program is obtained from its rendered visual content (using multi-modal LLMs like GPT-4o). Distilling the visual understanding of the symbolic graphics programs from vision encoder can better help LLMs to ground its textual and structured knowledge to the visual world.

---

> > > ### Comment · Reviewer_zWHc · 2024-11-22
> > >
> > > Thank you for your detailed reply and for running additional experiments. I appreciate the effort you have put into addressing my concerns and the novelty and value that the SGP-Bench benchmark brings to the community.
> > >
> > > However, based on the results provided, I remain unconvinced that the LLM demonstrates an "internal rendering" ability, as opposed to learning better adaptation strategies for vision DSLs (or potential shortcuts). This concern is particularly highlighted by the low performance on SGP-MNIST.
> > >
> > > Given these considerations, I have decided to maintain my rating at 6.

---

> > > > ### Author Response · Authors · 2024-11-23
> > > > **Thanks for the response!**
> > > >
> > > > We sincerely thank the reviewer for finding our work valuable and keeping the overall positive rating! We are also deeply appreciative of the reviewer's response for raising the concern. We hope our response can clarify the reviewer's remaining concerns.
> > > >
> > > > ---
> > > >
> > > > **Q1: However, based on the results provided, I remain unconvinced that the LLM demonstrates an "internal rendering" ability, as opposed to learning better adaptation strategies for vision DSLs (or potential shortcuts). This concern is particularly highlighted by the low performance on SGP-MNIST.**
> > > >
> > > > A1: This is an excellent question whether LLM demonstrates an ``internal rendering'' ability. In fact, this is one of the core questions that our benchmark seeks to answer. We believe that, instead of discussing whether LLM possesses the ability to do internal rendering, our work aims to study to what extent and to which degreee LLM can perform internal rendering and symbolic graphics program understanding.
> > > >
> > > > Our standard SGP-Bench evaluation set already shows that LLMs do have some ability to understand symbolic graphics programs. And moreover, the more powerful reasoning ability a LLM has, the better it performs on SGP-Bench. We would like to note **a key difference between SGP-Bench and SGP-MNIST: the abstraction level of the symbolic graphcis program**. One can observe that programs in SGP-MNIST (e.g. Figure 4 in the main paper) are typically composed of one operation and this operation is particularly long. Therefore, its abstraction level is quite low, which obviously constructs a more challenging case for LLMs to reason over. One of the valid hypotheses will be ``the difficulty of understanding a symbolic graphics program is highly dependent on its abstraction level''. Even in this extremely difficult case, OpenAI-o1 showcases strong reasoning/planning ability to find out the right answer (Figure 4 in the main paper). Therefore, it again echos our argument that how well LLMs perform on this task is informative for the indication of LLM's reasoning ability.
> > > >
> > > > We believe that the challenging cases provided by SGP-MNIST actually suggest many potential open questions, which, in our opinion, can inspire many more interesting future work in this field.

---

### Author Response · Authors · 2024-11-20
**General Response**

Dear Reviewers and ACs,

We sincerely thank all reviewers and ACs for spending time and efforts on our work. We are excited that all reviewers find our work interesting and our contribution novel. We are deeply appreciative of all the great suggestions from the reviewers. We carefully addressed each raised concern and hope that our rebuttal can clarify the concerns. If there is any remaining concern, we are more than happy to address them.

Best,

Authors

---

### Meta-Review · Area_Chair_zBoD · 2024-12-20

**Metareview:**

This paper studies exactly the question in the title: Can LLMs understand symbolic graphics programs. The goal is to see whether LLMs can answer semantics questions (of the rendered image) based on the raw program input. The authors introduce a new benchmark (SGP-Bench) covering SVG and CAD, evaluating on semantic understanding (annotated with GPT-4o with some human validation) as well as semantic consistency (does answers remain consistent with random translation/rotation). The authors evaluate existing LLMs as well as with instruction tuning to find that this can not only improve performance on SGP-Bench but also other general reasoning datasets. Finally, the authors demonstrate the difficulty for LLMs to understand SVGs through showing their poor performance on SGP-MNIST (classifying MNIST-like digits). There is unanimous decision from reviewers to accept the paper - the AC agrees that the paper proposes an interesting new benchmark, performs well-designed experiments, and the paper is well-written. Reviewers have brought up valuable points to improve the paper, including a discussion on limitations of the benchmark, connections to VLMs for solving these problems (besides it being used to generate annotations), potential training data leakage from captioned SVGs / other graphics programs in the pre-training set of existing LLMs. Also, multiple reviewers have brought up concerns on the 90% labeling accuracy for the questions - it would be good for the authors to also add this discussion in the main paper on question difficulty and any annotator variability.

**Additional Comments On Reviewer Discussion:**

Reviewers initially raised concerns about the value of the observations, data generation process, and potential shortcuts the LLMs could use to answer SGP-Bench questions. The authors were able to address most of the reviewer concerns during rebuttal, and all reviewers voted to accept the paper. I think the biggest concern is drawing conclusions on reasoning that could be contaminated by LLMs pre-trained on the internet that have already seen graphics programs, and adding additional experiments/discussions suggested by reviewers to address this point to the main paper would be great.

---

### Decision · Program_Chairs · 2025-01-22

Accept (Spotlight)